



# Secondary organic aerosol formation from smoldering and flaming combustion of biomass: a box model parametrization based on volatility basis set

Giulia Stefenelli[1], Jianhui Jiang[1], Amelie Bertrand[1,2,3,] Emily A. Bruns[1,a], Simone M. Pieber[1,4], Urs Baltensperger[1], Nicolas Marchand[2], Sebnem Aksoyoglu[1], André S. H. Prévôt[1], Jay G. Slowik[1], and Imad El Haddad[1]

[1] Laboratory of Atmospheric Chemistry, Paul Scherrer Institute (PSI), 5232 Villigen, Switzerland

[2] Aix Marseille Univ, CNRS, LCE, Marseille, France

[3] Agence de l'Environnement et de la Maîtrise de l'Energie 20, avenue du Gresillé – BP 90406, 49004 Angers Cedex 01, France

[4] Empa, Laboratory for Air Pollution and Environmental Technology, 8600 Dübendorf, Switzerland

[a] now at: Washington State Department of Ecology, Lacey, WA, USA

*Correspondence to*: Imad El Haddad (imad.el-haddad@psi.ch), Jay G. Slowik (jay.slowik@psi.ch) and Jianhui Jiang (jianhui.jiang@psi.ch)



**Abstract**

Box model simulations based on the volatility basis set (VBS) approach were used to assess secondary organic aerosol (SOA) precursors and volatility distributions from residential wood combustion. Emissions were sampled from three different residential stoves at different combustion conditions (flaming vs. smoldering-dominated), aging temperatures (-10°C, 2°C and

15°C), and emission loads, then exposed to hydroxyl (OH) radicals in a smog chamber. Primary emissions of SOA precursor compounds, organic aerosol and their evolution during aging in the smog chamber were monitored by a comprehensive suite of gas and particle instrumentation, including a proton transfer reaction time-of-flight mass spectrometer (PTR-TOF-MS) and a high resolution time-of-flight aerosol mass spectrometer (HR-ToF-AMS). SOA precursors were classified according to their chemical composition and the identification of the nature of the precursors revealed useful to better constrain model

parameters, in particular SOA production rates and molecular characteristics of the condensable gases formed. The general aim of the model was the determination of the parameters describing the volatility distributions of the oxidation products from the different chemical classes considered and their temperature dependence. Novel parameterization methods based on a genetic algorithm (GA) approach allowed estimation of precursor class contributions to SOA and evaluation of the effect of emission variability on SOA yield predictions. Significant differences were observed in the gas-phase composition between

smoldering and flaming emissions. Smoldering phase emissions were dominated by oxidized VOCs with less than six carbon atoms family ($OVOC_{c<6}$) while the flaming phase exhibited higher contributions by the single-ring aromatic hydrocarbon (SAH) and polycyclic aromatic hydrocarbon (PAH) classes. For both phases studied, cresol and phenolic species provided a major contribution to SOA formation. In combination with state-of-the-art mass spectrometry analysis, the model framework developed herein may be generalizable for other complex emissions sources, allowing determination of the contributions to

SOA of different precursor classes at a level of complexity suitable for implementation in regional air quality models.

**1. Introduction**

Atmospheric aerosols impact visibility, human health, and climate on a global scale (Stocker et al., 2013; World Health Organization, 2013). A thorough understanding of their chemical composition, sources, and processes is a fundamental prerequisite to develop appropriate mitigation policies. Laboratory experiments using smog chambers enable the detailed

examination of the gas-phase composition and aging of different emissions such as biomass smoke (e.g. Bruns et al., 2016; Bian et al., 2017), car exhaust (Gordon et al., 2014a ; b, Platt et al., 2017 ; Gentner et al., 2017; Pieber et al., 2018), aircraft exhaust (Miracolo et al., 2011; Kılıç et al., 2018), or cooking emissions (Klein et al., 2016). Results from these studies consistently show that the measured concentrations of secondary organic aerosol (SOA), formed upon oxidation and partitioning of the oxidized vapors, greatly exceed estimated concentrations based on the oxidation of volatile organic

compounds (VOCs) traditionally assumed to be the dominant SOA precursors (Jathar et al., 2012). The SOA formed from these chemically speciated VOCs is defined as traditional SOA (T-SOA) and is explicitly accounted for in chemical transport models (CTMs). However, Robinson et al. (2007) suggested that a significant fraction of the unexplained SOA is due to the





oxidation of lower-volatility organics, i.e. semi-volatile and intermediate volatility organic compounds (SVOC and IVOC, respectively), collectively referred to as non-traditional SOA (NT-SOA) precursors (Donahue et al., 2009).

In spite of its importance, incorporating NT-SOA into current organic aerosol (OA) models remains challenging without the identification and the quantification of the most important precursors (Jathar et al., 2012). For simplification purposes several

methods based on the volatility of the emissions and a volatility-based oxidation mechanism have been developed. Currently the volatility basis set (VBS) scheme is considered to be the most suitable approach to simulate the aging processes of non-speciated organic vapors (Donahue et al., 2006). The VBS scheme represents OA as a discrete volatility-resolved mass distribution. Reactions are described by the transfer of OA mass between volatility bins, thereby accounting for the contribution of non-traditional vapors to SOA formation without the need to incorporate explicit chemical mechanisms. Robinson et al.

(2007) proposed that SVOCs, IVOCs and their products react with hydroxyl radicals (OH) to form products that are an order of magnitude lower in volatility than their precursors. Pye and Seinfeld (2010) proposed a single-step mechanism for the non-speciated SVOCs, where the products of oxidation were two orders of magnitude lower in volatility than the precursors. They used SOA-yield (defined as SOA mass formed divided by reacted precursor mass) data for naphthalene as a surrogate for all non-speciated IVOCs, even though these are thought to be mainly branched and cyclic alkanes (Robinson et al., 2007, 2010;

Schauer et al., 1999). Both methods have been implemented in plume models as well as regional and global chemical transport models and have reduced discrepancies between measured and predicted SOA concentrations and properties (Shrivastava et al., 2008; Dzepina et al., 2009; Pye and Seinfeld, 2010; Jathar et al., 2011). However, considerable uncertainties remain in the relative contributions of non-traditional precursors to different emissions, their ability to form SOA and their reaction rate constants (Jathar et al., 2014a). Limitations in SOA modelling are also a direct consequence of limitations in measurements;

namely undetected or unidentified precursors and limited number of studies available investigating the influence of different parameters such as temperature, emission load, and combustion regimes. For instance, the overwhelming majority of smog chamber studies have been conducted under summer-time conditions (20-30°C), preventing the assessment of temperature effects on both SOA-producing reactions and the partitioning thermodynamics (Jathar et al., 2013).

Similar limitations apply to the consideration of emissions in models. Biomass combustion is a major source of gas and

particle-phase air pollution on urban, regional and global scales (Grieshop et al., 2009; Lanz et al., 2010; Crippa et al., 2013; Gobiet et al., 2014; Chen et al., 2017; Bozzetti et al., 2017). Globally, approximately 3 billion people burn biomass or coal for residential heating and cooking (World Energy Council, 2016), often using old and highly polluting appliances. Emissions from these devices are highly variable depending on fuel type and fuel moisture (McDonald et al., 2000; Schauer et al., 2001; Fine et al., 2002; Pettersson et al., 2011; Eriksson et al., 2014; Reda et al., 2015; Bertrand et al., 2017), and typically include

a complex mixture of non-methane organic gases (NMOGs), primary organic aerosol (POA), and black carbon (BC). Once emitted into the atmosphere, organic compounds can react with oxidants such as OH radicals, ozone ($O_3$) and nitrate radicals ($NO_3$). These reactions remain poorly understood, which greatly hinders the quantification of wood combustion SOA in ambient air. Bruns et al. (2016) investigated the SOA formation from the stable flaming phase of residential log wood combustion from a single stove in a smog chamber. They reported that T-SOA precursors included in models account for only



to 27% of the measured SOA whereas 84 to 116% was from NT-SOA precursors mainly consisting of polycyclic aromatic hydrocarbons from incomplete combustion (e.g. naphthalene) and cellulose and lignin pyrolysis products (e.g. furans and phenols, respectively). The estimated SOA concentrations were based on the literature SOA yields of single precursors, obtained from smog chambers experiments, and a good agreement was observed between predicted and measured SOA.

However, the method suffers from two drawbacks. First, the dependence of the yields on the organic aerosol loading and temperature was not considered. Second, although the relative contributions of different precursors to SOA were estimated, thermodynamic parameters for chemical transport models (CTMs) were not determined. Based on the same experiments, the lumped concentrations of the non-traditional volatile organic compounds were constrained in a box model (Ciarelli et al., 2017a). Improved parameters could describe the volatility distributions and the production rates of oxidation products from

the overall mixture of precursors present in biomass smoke. While this method is well suited for CTMs (Pandis et al., 2013; Ciarelli et al., 2017b), it does not provide any information about the contribution of the different chemical classes to the aerosol. Similar limitations are associated with the study of other emissions, e.g. fossil fuel combustion or evaporation (Jathar et al., 2013, 2014b). The development of models capable of simulating the contribution of the different chemical species to the aerosol at different conditions is especially important in the light of the current development of highly time resolved chemical

ionization mass spectrometry, capable of quantifying these products. To realize the full potential of the data acquired by this instrumentation, a modelling framework capable of predicting the production rates and the partitioning between the gas and the particle phase of the oxidation products from complex emissions is required.

Here, we extend the past analysis investigating the most recent smog chamber data of residential wood combustion based on 14 experiments performed in 2014-2015 under various conditions. Different experimental temperatures of the smog chamber

were investigated; namely -10°C, 2°C and 15°C. Three different stove types were tested, including conventional and modern residential burners. Different emission load and different hydroxyl (OH) radical exposure were examined. Moreover distinct combustion regimes were sampled across the different experiments for the first time, to investigate the secondary organic aerosol chemical composition and yields from flaming and smoldering emissions. Integrated VBS-based model and novel parameterization methods based on a genetic algorithm (GA) approach were developed to predict the contribution of the

oxidation products of different chemical classes present in complex emissions and to better explain the SOA formation process, providing useful information to regional air quality models. Overall, this study presents a general framework which can be adapted to assess SOA closure for complex emissions from different sources.

## 2 Methods

### 2.1 Smog chamber set up and procedure

Two smog chamber campaigns were conducted to investigate SOA production from multiple domestic wood combustion appliances as a function of combustion phase, initial fuel load, and OH exposure. These experiments were previously described in detail (Bruns et al., 2016; Ciarelli et al., 2017; Bertrand et al., 2017, 2018a) and are summarized here.





The emissions were generated by three different logwood stoves for residential wood combustion: stove 1 manufactured before 2002 (Cheminés Gaudin Ecochauff 625), stove 2 fabricated in 2010 (Invicta Remilly) and stove 3 (Avant, 2009, Attika). For each stove three to four replicate experiments were performed with a loading of 2-3 kg of beech wood having a total moisture content ranging between 2 and 19%. The fire was ignited with 3 starters made of wood wax, wood shavings, paraffin and

natural resin. The starting phase was not studied. In total, 14 experiments were performed, consisting of two experiments at -10°C, seven experiments at 2°C and five experiments at 15°C. These experiments cover the typical range of European winter temperatures and are summarized in Table 1.

The experiments were performed in a flexible Teflon bag of nominally 7 but typically about 5.5 $m^3$ equipped with UV lamps (40 lights, 90–100W, Cleo Performance, Philips, wavelength $\lambda < 400$ nm) enabling photo-oxidation of the emissions (Platt et

al., 2013; Bruns et al., 2015). The chamber is located inside a temperature-controlled housing. Relative humidity was maintained at 50% and three different temperatures were investigated. Emissions from the stoves were sampled from the chimney into the chamber through heated (140°C) stainless-steel lines to reduce the loss of semi-volatile compounds. An ejector dilutor was installed (Dekati Ltd, DI-1000) to dilute emissions in the chamber by a factor of 10 before sampling. The sample injection lasted for approximately 30 minutes for each experiment, and were followed by an injection of 1 μL d9-

butanol (98%, Cambridge Isotope Laboratories), which was used to estimate the OH exposure as described in Section 3.1.1 (Barmet et al., 2012). The chamber was then allowed to equilibrate for 30 minutes to ensure stabilization and homogeneity and to fully characterize the primary emissions before aging. OH radicals were produced by UV irradiation of nitrous acid (HONO) injected after chamber equilibration, generated as described in Taira and Kanda (1990), reaction of diluted sulfuric acid ($H_2SO_4$) and sodium nitrate ($NaNO_3$) in a gas flask, and introduced into the chamber and brought into the gas-phase

flushing with a carrier gas with a flow rate of 1 L min$^{-1}$. The smog chamber was then irradiated with UV lights for approximately 4 hours to simulate atmospheric aging.

Before and after each experiment, the smog chamber was cleaned with humidified pure air (100% RH) and $O_3$ (1000 ppm) under irradiation with UV lights for at least 1 hour, followed by flushing with pure dry air for at least 10 hours. The background particle- and gas-phase concentrations were then measured in the clean chamber.

The total amount and composition of the emissions depends on the oxygen supply, temperature, the fuel elemental composition, and combustion conditions, which can be broadly classified as flaming or smoldering (Koppmann et al., 2005; Sekimoto et al., 2018). Flaming combustion occurs at high temperature and consists of volatilization of hydrocarbons from the thermal decomposition of biomass leading to rapid oxidation and efficient combustion, producing $CO_2$, water and black carbon (BC). Instead, smoldering combustion is flameless and can be initiated by weak sources of heat and results in less efficient

combustion of fuel, leading to gas-phase products (mainly CO, $CH_4$ and volatile organic compounds). During the experiments in Set2, we focused on flaming emissions, while in Set1 combined flaming and smoldering emissions were studied in the smog chamber.



## 2.2 Instrumentation

We characterized the emissions with a suite of gas- and particle-phase instrumentation. Non-methane volatile organic compounds (VOCs) were measured by a proton transfer reaction time-of-flight mass spectrometer (PTR-ToF-MS 8000, Ionicon Analytik). A detailed description of the instrument can be found in Jordan et al. (2009). The PTR-ToF-MS was

operated under standard conditions (ion drift pressure of 2.2 mbar and drift intensity of 125 Td) in $H_3O^+$ mode, allowing the detection of VOCs with a proton affinity higher than that of water. For quantification, when known individual reaction rate constants were used (Cappellin et al., 2012), otherwise a value of $2 \times 10^{-9}$ $cm^3$ $s^{-1}$ was assumed. The effectively rate constants applied to both Set1 and Set2 can be found in Bruns et al., (2017). Data were analyzed using the Tofware software 2.4.2 (PTR module as distributed by Ionicon Analytik GmbH, Innsbruck, Austria) running in Igor Pro 6.3.

Non-refractory primary and aged particle composition was monitored by a high resolution time-of-flight aerosol mass spectrometer (HR-ToF-AMS, Aerodyne Research Inc.) (DeCarlo et al., 2006). The HR-ToF-AMS is described in detail elsewhere (Bruns et al., 2016; Bertrand et al., 2017) and summarized here. The instrument was operated under standard conditions (temperature of vaporizer 600°C, electronic ionization (EI) at 70eV, V mode) with a temporal resolution of 10 seconds. Data analysis was performed in Igor Pro 6.3 (Wave Metrics) using SQUIRREL 1.57 and PIKA 1.15Z assuming a

collection efficiency of 1. The O:C ratio was determined according to Aiken et al. (2008).

Black carbon (BC) was derived from the absorption coefficient measured with a 7-wavelength aethalometer (Magee Scientific aethalometer model AE33). The corresponding mass concentration of equivalent BC (eBC) was thus converted from the absorption coefficient measured with a time resolution of 1 minute at a wavelength of 880 nm (Drinovec et al., 2015).

The particle number concentration and size distribution (16 to 914 nm) were provided by a scanning mobility particle sizer

(SMPS, consisting of a custom-built differential mobility analyzer (DMA) and a condensation particle counter (CPC 3022, TSI)) with a time resolution of 5 minutes. Supporting gas measurements included a $CO_2$ analyzer (LI-COR), a $CH_4$, a total hydrocarbon (THC) monitor (flame ionization detector, THC monitor Horiba APHA-370), and NO and $NO_2$ ($NO_X$ analyzer, Thermo Environmental) monitors.

## 3. Data analysis

The data analysis entails three steps detailed in this Section: The first sub-section describes the determination of the amount of oxidized VOCs in the chamber. The second sub-section details the determination of the amount of SOA formed in the chamber. The last sub-section describes the box model used for the parameterization of SOA formation from the VOCs.

### 3.1 VOC loss in the chamber

In the chamber, VOCs were oxidized to several oxidation products, referred to as oxidized VOCs (CG, condensable gases) in

the following analysis. According to their volatility, these products may remain in the gas phase or partition to the particle phase, thereby contributing to SOA formation.



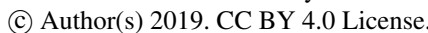 

We described the change in any VOC concentration over time as a combination of its loss and production as follows:

$$\frac{d[VOC]}{dt} = P - \sum k_{dil} * [VOC] + k_{OH} * [OH] * [VOC] + k_{other} * [VOC] \qquad (1)$$

Here, $P$ corresponds to the production of a VOC in the chamber, e.g. from the oxidation of other primary VOCs. $k_{dil}$ is the
dilution rate constant in s$^{-1}$. $k_{OH}$[OH] [VOC] in molec$^{-1}$ cm$^3$ s$^{-1}$ represents the consumption rate due to oxidation by OH, where
$k_{OH}$ is the reaction rate constant and [OH] is the OH concentration. $k_{other}$[VOC] in molec$^{-1}$ cm$^3$ s$^{-1}$ is the loss rate of VOC by
other processes, where $k_{other}$ is the reaction rate constant in s$^{-1}$. The loss of some VOCs could not be explained by their
reaction with OH and dilution alone for the Set1, so we added this additional term which is discussed after the first two
processes are constrained. We considered primary VOCs that exhibited a clear decay with time to be strictly of primary origin,
and hence neglected their production from other VOCs (i.e. $P = 0$). This assumption signifies that the yields estimated under
our conditions are upper limits. In reality, the detection of aromatic hydrocarbons (e.g. single-ring aromatic hydrocarbons,
SAHs and polycyclic aromatic hydrocarbons, PAH) by the PTR-ToF-MS may be affected by the interference due to
fragmentation during ionization of their oxidation products (Gueneron et al., 2015). On the other hand, directly emitted
oxygenated aromatics could be themselves the oxidation products of aromatic hydrocarbons and their production may continue
during the experiment. However, the assumption of $P = 0$ does not introduce a significant error for most VOCs with significant
primary emissions, because the observed VOC decay was consistent with their OH reaction rate constant for Set 2 as
demonstrated by Bruns et al. (2017) for the +15°C conditions. In the following, we describe the processes governing the
changes in the VOC concentrations in the chamber and the approaches adopted for the determination of the different parameters
in Eq. (1).

**3.1.1 Reaction with OH radical**

The OH exposure, that is the integrated OH concentration over time, was estimated based on the differential reactivity of two
VOCs. Specifically, we used d9-butanol (fragment at mass to charge ratio $m/z$ 66.126, [C$_4$D$_9$]$^+$) and naphthalene (fragment at
$m/z$ 129.070, [C$_{10}$H$_8$]H$^+$). These compounds are selected because they can be unambiguously detected (no isomers or
interferences, high signal-to-noise), are not produced during the experiment, and have OH reaction rate constants that are
precisely measured and significantly different from each other. The OH exposure can be expressed as follows:

$$OH\ exposure = \left( \frac{ln\left(\frac{d9-butanol}{naphthalene}\right)_0 - ln\left(\frac{d9-butanol}{naphthalene}\right)_t}{k_{OH,but} - k_{OH,naph}} \right) \qquad (2)$$



where (d9-butanol/naphthalene)$_0$ is the ratio between these compounds at $t = 0$ (before lights were turned on), (d9-butanol/naphthalene)$_t$ is the ratio measured at time $t$, and $k_{OH,but}$ and $k_{OH,naph}$ are the OH reaction constants of d9-butanol and naphthalene, respectively ($k_{OH,but}=3.14 \times 10^{-12}$ cm$^3$ molec$^{-1}$ s$^{-1}$ and $k_{OH,naph} = 2.30 \times 10^{-11}$ cm$^3$ molec$^{-1}$ s$^{-1}$) (Atkinson and Arey, 2003).

For Set1, the OH exposure at the end of each experiment ranged between 5 and $8 \times 10^6$ molec cm$^{-3}$ h, corresponding to approximately 5-8 hours in the atmosphere (given global average and typical wintertime OH concentrations of $1 \times 10^6$ molec cm$^{-3}$). For Set2, higher OH exposures were reached (3 to $7 \times 10^7$ molec cm$^{-3}$ h at the end of each experiment, corresponding to 2-3 days in the atmosphere). This is likely because both sets of experiments utilized a similar HONO molar flow (and thus similar OH production rate), but higher VOC concentrations were reached in Set1, which could possibly have resulted in a

higher OH sink. We calculated the OH concentration, [OH] in Eq. (1), numerically as $d$(OH exposure)/$dt$.

### 3.1.2 Smog chamber dilution

For Set 2, dilution was dominated by the constant injection of HONO to the chamber and accounted for as described in Bruns et al. (2016). For Set1 the dilution rate of primary VOCs in the chamber was calculated as follows. The integrated dilution over time, $K_{dil}$, was determined as the ratio between the d9-butanol concentration corrected for the reaction of d9-butanol with

OH and the d9-butanol concentration at $t = 0$ ([d9-butanol]$_0$):

$$K_{dil} = \frac{[d9 - butanol] * e^{k_{OH,but}*OH\ exposure}}{[d9 - butanol]_0}$$

(3)

Figure S1 shows calculated values for $K_{dil}$ as a function of time. Increasingly high dilution at the end of these experiments (Fig S1), between 20 and 35% (i.e., a dilution ratio of 0.8 to 0.65), is a result of constant injection of HONO and excessive sampling at high rates, resulting in inputs from laboratory air (likely through leaks in the Teflon bag or the connections of chamber inlets

and outlets). The inputs from the HONO pure carrier gas and of laboratory air are roughly comparable (about 12 vs. 8-23% at the end of the experiment). Besides dilution, the effect of both inputs on the gas phase chemical composition is negligible. Dilution rates are non-linear, increasing as the experiment progresses due to continuous dilution within a decreasing chamber volume. The dilution rate constant, $k_{dil}$, in Eq. (1), is the differential of $K_{dil}$ over time.

### 3.1.3 Losses by other processes

To corroborate the estimated OH exposures and dilution rates, we examined the loss of prominent VOCs with known reaction rate constants against OH. For Set2 and as demonstrated by Bruns et al. (2017), the VOC decay is consistent with their estimated loss based on their dilution and reaction with OH. By contrast, for Set1, the decay of some VOCs could not be solely explained by their reaction with OH and dilution, suggesting additional reactions with oxidants other than OH as discussed below. The VOC total consumption by this process ($\int k_{other}$ [VOC]) was estimated as the difference between the total





measured decay of the VOC of interest and the fraction consumed by both dilution and oxidation by OH radicals. The reaction rate constants for several precursors towards OH ($k_{OH}$) are not available in the literature. In addition, many fragments may have several isomers, each of which associated with different rate constants. Effective rate constants for all precursors considered were estimated from their decay in Set2, where the combination of OH reaction and dilution fully explained the

decay of VOCs with known OH reaction rate constants.

### 3.1.4. Precursor classification

A common set of 263 ions were extracted from the PTR-ToF-MS. Among these ions, 86 showed a clear decay with time and were thus identified and selected as potential SOA precursors. Previous work based on Set2 experiments showed that the PTR-ToF-MS measures the most important SOA precursors, which explained the measured SOA mass within 40% uncertainty and

without systematic bias (Bruns et al., 2016). These 86 ions were grouped into 6 classes: furans, single-ring aromatic hydrocarbons (SAH), polycyclic aromatic hydrocarbons (PAH), oxygenated aromatics (OxyAH) and organic compounds containing more or less than 6 carbon atoms ($OVOC_{c\geq6}$, $OVOC_{c<6}$, respectively) (Table S1). We believe these classes capture the dominant fraction of SOA mass, although we cannot rule out losses in the PTR-ToF-MS inlet or small contributions from other precursors such as alkanes. The identification of each compound was supported by previous publications (McDonald et

al., 2000; Fine et al., 2001; Nolte et al., 2001; Schauer et al., 2001; Stockwell et al., 2015) including gas chromatography-mass spectrometry (GC-MS) analysis when available.

### 3.2 Calculation of the OA mass in the chamber

The total organic aerosol measured by the HR-ToF-AMS was corrected for particle losses in the chamber due to gravitational and diffusional deposition. To assess the total wall losses due to both processes, we assumed that the condensable vapors

partition only to the suspended aerosols but not to the wall.

Assuming that black carbon is inert in the chamber, it was possible to use its decay to estimate the particle loss to the walls. The aerosol attenuation measured at 880 nm (at the end of each experiment, ~4 hours) with an aethalometer was used to estimate the particle loss rate to the wall. This attenuation is proportional to the eBC mass concentration and within uncertainties independent of the aging extent, as demonstrated in Kumar et al. (2018). Using eBC as a tracer, we inherently

assumed that eBC and OA were internally mixed and homogeneously distributed over the aerosol size range. The decay of eBC due to both dilution and deposition onto the chamber walls was parametrized as follows:

$$\frac{d[\text{eBC}]}{dt} = -k_{dil}\,[\text{eBC}] - k_{wall}\,[\text{eBC}]$$

(4)

where $k_{wall}$ is the first order wall loss rate used to correct the measured OA concentration for wall losses, ranging between 4 and $8 \times 10^{-5}$ s$^{-1}$, and $k_{dil}$ is the dilution rate determined above.



The wall loss corrected organic aerosol, $OA_{WLC}$, was calculated using Eq. (5):

$$OA_{WLC}(t) = OA(t) + \int_0^t k_{wall} * OA(t) * dt$$

(5)

where $OA(t)$ is the measured organic aerosol concentration in µg m$^{-3}$. The total OA present in the chamber was estimated as the suspended OA concentration measured by the HR-ToF-AMS plus the estimated OA lost to the wall. This concentration was directly compared to the condensable gases (CG) concentration estimated according to Eq. (A8) presented in the Appendix A. As mentioned, our approach did not take into account the losses of precursor vapors or their oxidation products in the gas phase onto the walls. We note that these processes were unlikely to have a substantial effect on the precursors considered, which were largely highly volatile species, even at lower temperature. Based on the calculation of the equilibrium constant of semi-volatile species on the walls by Bertrand et al. (2018b), we estimated that at 293 K the fraction of these compounds absorbed on the walls is <5%. Meanwhile, the walls could indeed act as a sink for the semi-volatile oxidation products. This effect was not taken into account in the current study, but we expect that it was minimized under our conditions, by the high OA concentration in the chamber and rapid production rates (Zhang et al., 2014; Nah et al., 2017).

### 3.3. Modelling SOA formation

The general aim of the model is the determination of the parameters describing the volatility distributions of the oxidation products from different precursor classes and their temperature-dependence. A simplified schematic of the modeling framework is described in Fig. 1. It consists of 1) a box model that describes the partitioning of the condensable gases generated through oxidation, 2) the model input parameters obtained from the smog chamber, 3) the model output parameters and 4) the model optimization based on a genetic algorithm (GA). Each of these parts is described in the following sections.

### 3.3.1 Box model

We assume the partitioning of CG between the gas and the particle phases to obey Raoult's law (Strader et al., 1999), where the aerosol can be described as a pseudo-ideal organic solution, of SOA and POA species. The volatility basis set (VBS, implemented by Koo et al., 2014 in the Comprehensive Air quality Model with eXtensions, CAMx), was used to classify the oxidation products of the different precursors into surrogates with different volatility, distributed into discrete logarithmically spaced bins (Donahue et al., 2006).

We considered the most basic mechanism by which SOA may form. That is, the oxidation products from the different precursor classes described above instantaneously partition into the condensed phase depending on their volatility. No additional reactions in the gas or particle phase were considered (e.g. reaction with oxidants, photolysis or oligomerization). In addition, we neglected the contribution of primary oxidation products of the gas-phase semi-volatile species (co-emitted with POA) compared to the VOCs detected by the PTR-ToF-MS, based on the findings of Bruns et al. (2016) and Ciarelli et al. (2017).



Finally, we considered the species in the gas and the particle phase to be permanently at equilibrium, as condensation is expected to be faster than oxidation (time scales for oxidized vapours condensation < 1 minute assuming no particle phase diffusion limitations, Bertrand et al., 2018b). While including additional processes in the model is feasible, this would result in a significantly higher-dimensional parameter space, which cannot be unambiguously inferred from the present data. We

consider that without supportive data, e.g. chemically resolved characterization of the particle phase species, such reactions could not be well constrained or even deduced from structure activity relationships, given the many unknowns in complex emissions. Therefore, such simplified scheme of SOA formation from complex emissions may be compared in the future with chemically resolved data to help the identification of additional mechanisms that were not considered here.

The derivation of the thermodynamic equations governing SOA formation from precursors, implemented in the box model, is

detailed in Appendix A and only a brief description of the model principles is given here. In the following, let $i$ and $j$ be the indices for the different volatility bins and precursor classes, respectively. The model determines the molar distribution of the oxidation products from different precursor classes in the different volatility bins, $Y_{i,j}$, together with the compounds' enthalpy of evaporation, $\Delta H_{vap\ i,j}$. The latter describes the temperature dependence of the oxidation products' effective molar saturation concentration, $x^*_{i,j}$. For this, the model iteratively solves Eq. (A6) and (A7) at every experimental time step (time resolution of

10 seconds) for all experiments, to retrieve the surrogate molar concentrations in the particle phase, $x_{i,j}|_p$ and the total surrogates' molar concentration in the condensed organic phase, $x_{OA}$ (see Appendix A).

### 3.3.2 Model inputs

The model uses as main inputs the molar concentrations of the condensable gases from different precursors (in total $n = 6$ precursor classes) in both phases, $x_j|_{g+p}$. The latter is derived from the consumption rates of $VOC_j$ determined by the PTR-

ToF-MS, by numerically integrating Eq. A8. $x_j|_{g+p}$ is related to the concentrations of the different surrogates from a precursor class $j$ in different volatility bins, $x_{i,j}|_{g+p}$ (Eq. A6), through their yields, $Y_{i,j}$, according to Eq. A9. The number of volatility bins, $m$, is set to 6, approximately corresponding to the following mass saturation concentrations: $\overrightarrow{C^*_{i,j}}(\mu g\ m^{-3}) = \{\ 10^{-1}; 10^0; 10^1; 10^2; 10^3; 10^4 \}$.

In addition to $x_j|_{g+p}$, the model needs as inputs $x_i^{OM_p}|_{p+g}$, the molar concentration of primary organic matter from a volatility

bin $i$ in both gas and particle phase (Eq. A7). $x_i^{OM_p}|_{g+p}$ is inferred from the measured POA concentrations injected in the chamber at the beginning of the experiment and using the volatility distribution function of wood combustion emissions in May et al. (2013). It is assumed constant with aging. The computation of the fraction of $POA_i$ in the condensed phase is similar to that for SOA species in Eq. (A6).

The secondary surrogates' elemental composition ($C\#_{i,j}$, $O\#_{i,j}$ and $H\#_{i,j}$) is also used as model inputs to compute the

surrogates molecular weight, $MW_{i,j}$, required for $C_{OA}$ calculations (see Section 3.3.3). A single $C\#_{i,j}$ value is calculated per chemical class, based on the average $C\#_j^{VOC}$ of the respective precursor class, and considering $C\#_{i,j} = C\#_j^{VOC} - \Delta C$, where



$\Delta C$ is the average loss in carbon due to fragmentation during the oxidation of precursors from all classes. $\Delta C$ is determined by systematically changing its value in multiple model runs and selecting the value that explains best the observed O:C ratios (see Fig. 8b). Likewise, a single $H\#_{i,j}$ value is assumed per chemical class, considering that $H\#_j/C\#_j$ equals $H\#_j^{VOC}/C\#_j^{VOC}$. Finally, $O\#_{i,j}$ is constrained by the $C\#_j$ and the surrogate volatility ($C_{i,j}^*$) based on the simplification of the SIMPOL model

5 (Pankow and Asher, 2008), provided by Eq. (3) in Donahue et al. (2011). Based on this relationship, $O\#_{i,j}$ increases with decreasing $C\#_j$ and $C_{i,j}^*$. The O:C ratio of primary emissions is constrained in the model to the measured O:C in the beginning of each experiment, by setting $C\#^{OM_p}$ and calculating $O\#_i^{OM_p}$ using the same methodology as for $O\#_{i,j}$. The resulting $C\#^{OM_p}$ and $O\#_i^{OM_p}$ and the corresponding primary organic matter molecular weight, $MW_i^{OM_p}$, as well as $C\#_j$, $O\#_{i,j}$, $H\#_j$ and $MW_{i,j}$ are reported in Table S4.

10 ### 3.3.3 Model outputs

The model provides the $Y_{i,j}$ and $\Delta H_{vap\ i,j}$ parameters. To reduce the model's degree of freedom we consider a single $\Delta H_{vap}$ for all surrogates from different chemical classes in different volatility bins. $Y_{i,j}$ is considered to follow a kernel normal distribution as a function of $\log C^*$, $Y_{i,j} \sim N(\mu_j, \sigma)$, where $\mu_j$ is the median value of $\log C^*$ and $\sigma$ is the standard deviation. This step (1) insures positive $Y_{i,j}$ parameters, (2) significantly reduces the model's degree of freedom and (3) allows constraining

15 the total concentration of surrogates from a certain chemical class: $\sum_i^m Y_{i,j} = 1$.

The set of $Y_{i,j}$ and $\Delta H_{vap\ i,j}$ parameters are determined by minimizing the sum of mean bias (MB) and the root mean square error (RMSE) between modelled mass concentrations of the particulate organic phase, $C_{OA}$, calculated using Eq. (6) and concentrations measured by the AMS.

$$C_{OA} = \sum_j^n \sum_i^m MW_{i,j} x_{i,j}|_p + MW_i^{OM_p} x_i^{OM_p}|_p \tag{6}$$

The model fitted to the measured $C_{OA}$ was also validated by external AMS measurements of the O:C ratio determined through high resolution analysis. The modelled O:C ratio was calculated at every experimental time step as follows:

$$O:C = \frac{\sum_j^n \sum_i^m O\#_{i,j} x_{i,j}|_p + O\#_i^{OM_p} x_i^{OM_p}|_p}{\sum_j^n \sum_i^m C\#_{i,j} x_{i,j}|_p + C\#_i^{OM_p} x_i^{OM_p}|_p} \tag{7}$$



### 3.3.4 Model optimization

For the model optimization, we used a genetic algorithm (GA), a metaheuristic procedure inspired by the process of natural selection, including mutation, crossover and selection, to efficiently generate high-quality solutions to optimize problems. A population of 50 different sets of model parameters ($\mu_j$, $\sigma$ and $\Delta H_{vap}$) was considered for each GA generation. The termination criteria was no improvement in the fitness function, MB and RMSE between measured and modelled $C_{OA}$, after 50 generations, with a maximum of 500 total iterations allowed. The GA calculations were performed using the package "GA" for R (Scrucca et al., 2017). A bootstrap method was then adopted to quantify the uncertainty in the constrained parameters.

## 4. Results and discussion

### 4.1 Comparison of primary emissions across experiments

A larger amount of primary VOCs was emitted into the chamber in Set1, with concentrations ranging from 950 to 7860 µg m$^{-3}$, while in Set2 the primary VOCs concentrations ranged from 300 to 1360 µg m$^{-3}$. In the same way, the measured OA at the beginning of each test (POA) ranged from 10 to 180 µg m$^{-3}$ and from 9 to 22 µg m$^{-3}$ for Set1 and Set2, respectively (Table 1). The VOC composition for Set1 and Set2 is summarized in Fig. 2 showing the mean PTR-ToF-MS mass spectra (Fig. 2a, b), the relative contributions of the different compounds for different datasets (Fig. 2c) and the variability in composition among all experiments (Fig. 2d). Set1 shows higher relative contributions of furans and OVOC$_{c<6}$ (Fig. 2a), while the contributions of PAH, SAH and OxyAH are higher in Set2 (Fig. 2b). The OxyAH compounds, mainly methyl and methoxy-phenols, are produced by lignin pyrolysis (Fine et al., 2001) while furans are formed from cellulose pyrolysis (Mettler et al., 2012). The majority of the compounds differ among datasets and the most significant difference estimated through the *p* value (probability associated with a student *t*-test) occurs for the OVOC$_{c<6}$, which is about a factor of 6 higher than the significance threshold ($p = 0.05$) for Set1. In order to investigate the similarity between all experiments a Spearman correlation matrix was calculated. Experiments from Set1 appear to be consistently similar to each other while the experiments from Set2 are significantly different among each other in terms of composition of the primary emissions. A possible reason for such discrepancy was the difficulty in injecting flaming emissions without any significant smoldering contribution for Set2. This hypothesis is supported by the strong similarity between precursor compounds measured in some experiments supposed to represent flaming phase emissions only but apparently included some smoldering as well (9, 12, 13) with experiments from Set1 (1-7). Figure S2 reports the relative contributions of primary VOCs (before photo-oxidation) for all compound classes for the fourteen experiments. Note that these trends are not correlated with the modified combustion efficiency (MCE), reported in Table 1, defined as $CO_2/(CO+CO_2)$, which was constant at 0.97 g g$^{-1}$, for Set2, but ranged from 0.8 g g$^{-1}$ to 0.91 g g$^{-1}$ for Set1.

Figure 3 shows the contribution of each class of precursor compounds and of the primary semi-volatile organic matter (OM$_P$) to the total primary emissions. OM$_P$ is the total organic matter in the semi-volatile and low-volatile range in the particle and the gas phase (saturation concentration < 1000 µg m$^{-3}$ at 298 K) (see Section 3.3.2). Overall, the highest average relative



contributions are related to $OVOC_{c<6}$ followed by furans and OxyAH but we also note an average large contribution by SAH for Set2. The two sets of experiments investigated clearly show different primary composition of emissions in terms of dominant contributions; in Set1 $OVOC_{c<6}$ and OxyAH dominate by far the total primary emissions while in Set2 the main species influencing the total primary emissions are OxyAH, $OVOC_{c<6}$ and SAH with roughly similar contribution (see Fig.

3b). Moreover, the calculated averaged $OM_p$/VOCs ratios are around 0.05 and 0.03 for Set2 and Set1, respectively.

VOCs undergo oxidation during atmospheric aging to form a complex mixture of products, some of which remain in the gas phase while others have sufficiently low volatility to partition to the particle phase. The consumption of the different VOC classes over time is shown in Fig. 4a, b for Set1 and Set2 respectively. The general trends manifest that PAH and OxyAH are the most reactive classes, exhibiting an average consumption of up to 80% at the end of the experiments (after ~4 hours of

aging) while for both the datasets SAH appears to be the least reactive class (with an average consumption between 10 and 20% at the end of experiments). Relevant compounds in the latter class are benzene ($C_6H_6$), toluene ($C_7H_8$) and xylene ($C_8H_{10}$), their slow reactivity is consistent with literature reaction rate constants toward OH, from the NIST database (NIST chemistry WebBook, 2018), of $1.22 \times 10^{-12}$, $6.13 \times 10^{-12}$ and $7.51 \times 10^{-12}$ ($cm^3$ $molec^{-1}$ $s^{-1}$), respectively.

The two datasets differ for the OH dose; we observe in Set2 (Expt. 8-14) an overall higher consumption of all precursor classes

due to the higher OH dose (representing a longer aging time in an ambient atmosphere ). PAH shows the highest reactivity followed by OxyAH while for Set1 (Expt. 1-7) the fastest class of compounds to react is the OxyAH followed by PAH and $OVOC_{c<6}$. Moreover, despite of the lower OH exposure reached for Set1 the consumption of OxyAH and furans is substantially higher at comparable exposure levels.

In the same way, we also observe a higher SOA production for Set1 compared to Set2 at comparable OH exposure. The

SOA/POA ranges between 2 and 6, similar to ratios observed in previous studies.

Overall the same chemical classes appear to behave differently across the different sets of experiments. Such an inconsistency in behavior is either due to differences in the chemical composition within the same class or due to additional reactivity occurring in Set1.

To investigate the chemical differences within the same class of compounds across different experiments, Table 2 reports the

average reaction rate constants against OH of the different chemical classes calculated at the beginning of each experiment following Eq. (8).

$$\bar{k}_{OH_{j,k}} = \sum_i k_{OH_{c,j}} \times \frac{VOC_{c,j,k}}{VOC_{j,k}} \qquad (8)$$

Here, $c$ represents the single compound, $j$ the family and $k$ the experiment. $k_{OH}$ is the reaction rate constant toward OH and

VOC refers to the primary VOCs.

OH reaction rate constants ($k_{OH_{c,j}}$) for each compound were calculated from Set2 only (Fig. 5), whereas the decay of the precursors contributing most to SOA formation during aging was compared with the expected decay based on literature. The




good agreement indicates that for these experiments the consumption of the precursors was dominated by OH (Bruns et al. 2017). The average OH reaction rate constants ($k_{OH_{c,j}}$) are reported in Table S3. They are determined for each precursor class and calculated with a first order exponential fitting on the precursors' decay curves previously corrected for dilution.

The average reaction rate constants per family ($\bar{k}_{OH_{j,k}}$) are similar among the same families for different experiments

suggesting that the variable behavior of the chemical classes across different experiments was due to differences in the reactive environment rather than a different chemical composition within a given class.

As introduced in Section 3.1.3, the total measured decay of VOCs in Set1 could not be fully explained by dilution and reactivity against OH, suggesting the presence of an additional loss process. To assess the remaining oxidation processes, $k_{OH}$ values were used to estimate the missing loss process for Set1 according to Eq. (1). In this way the consumed fraction due to OH

chemistry, dilution in the chamber and the additional reactivity was calculated for each VOC compound family and is shown in Fig. 6. The additional reactivity appears to contribute to the total precursor consumption for most of the classes, with particular relevance for the OxyAH and PAH classes. This is in contrast with the calculated loss for the Set2 (Fig. S3) where the dominant consumption is due to OH.

One possible hypothesis is that the remaining loss process might be due to reaction with the nitrate radical ($NO_3$), which

absorbs in the visible region (~500-650 nm) and thus is not efficiently photolyzed by the black lights used here (Reed et al., 2016). Fig. S4 shows that for Set1 the $NO_3$ reaction rate constants ($k_{NO3}$) for compounds found in the NIST database (NIST chemistry WebBook, 2018) (see Table S1) are well correlated with the amount reacted, making nitrate chemistry a likely loss process (Schwantes et al., 2018). We note that while the OH production rate in the chamber is similar for the two sets of experiments, given the same injection rate of HONO, the OH total reactivity is significantly higher for Set 1, because of the

injection of ~5 times higher VOC concentrations. As a result, the OH concentration is around $1.5 \times 10^6$ versus $10 \times 10^6$ molec cm$^{-3}$ for Set1 and Set2, respectively, increasing the availability of VOCs for consumption by other processes during Set1.

## 4.2 Model evaluation

As previously mentioned, the reaction of the VOCs produces oxidized VOCs (CG, condensable gases) which can consequently partition between the gas and particle phases. Their concentration, estimated by accounting for the production minus the loss

in the chamber, as described in Eq. (1), was used as input for the box model with the volatility basis set (VBS) scheme. Figure S5 shows the volatility distributions of different precursor classes for Set1 and Set2 to assess the influence of different processes occurring in different experiments.

We note that for Set1 the lower volatility bins exhibit higher contributions compared to Set2 but still within two standard deviations, such that it is difficult to distinguish statistically different yields for most of the cases. Hence, the total volatility

distribution was used to calculate the mass yield instead of a specific one for each dataset (Fig. 9). Mass yields calculated for the specific datasets Set1 and Set2 are reported in Fig. S6.



Fig. 7 shows the modelled and measured OA mass for all fourteen experiments, where Set1 accounts for both OH and $NO_3$ chemistry while Set2 includes OH chemistry only. The modelled OA is divided into POA and 6 SOA classes attributed to the respective precursor classes. Overall, the model performance is satisfactory, although in general the final OA concentration is slightly overpredicted while the initial production rate is underpredicted. Most of the SOA is attributed to furans (30.8%),

OxyAH (19%) and $OVOC_{c<6}$ (12.5%) for Set1 while for Set2 there is a generally lower contribution from $OVOC_{c\geq6}$ (7.8%) and a higher contribution from PAH (12%), especially for experiments 08, 13 and 14. The mean bias between measured and modelled OA averaged over all experiments is -7.2 µg m$^{-3}$ which corresponds to ~ 15 % on average.

## 4.3 Investigation of OA chemical and physical properties

Comparisons between measured and modelled O:C ratios are reported in Fig. 8. The oxidation products elemental composition
based on which the modelled O:C ratio is calculated are presented in Table S4. The oxidation products' carbon numbers that explained best the observed O:C ratio corresponds to a set $\Delta C$ of 0.6 (Fig. S7). There is a general increase in the O:C ratio with time. Model and observations match in terms of average O:C ratio for each experiment but the temporal evolution of the ratio is not well predicted suggesting that there are additional processes that are not taken into account in the model. As the model was initiated using the measured POA O:C ratio at OH exposure equal to zero, an agreement between model and
measurements can be observed at this time of the experiment. We do not find any systematic correlation of the bias with chamber conditions except for lower concentrations where experiments exhibit higher O:C ratios at the end. Overall for experiments conducted at lower temperature (-10°C) the model tends to overestimate the O:C while for higher temperature experiments (15°C) the model clearly under-predicts the ratio upon aging especially at the end of the experiments indicating the presence of compounds with a higher number of oxygen (lower number of carbon) than predicted.

Fig. 9 shows mass yield curves for each class of compounds, in comparison with mass yields of several single compounds from literature (Table S2). The single compounds were selected according to their presence in the current study and compared to the respective chemical family. The reported published yields were discriminated according to experimental $NO_x$ regimes, as low NOx conditions generally lead to higher OA yields. Our study is more likely representative of a high NOx regime and thus assesses the SOA forming potential for this atmospherically relevant condition. For the model and measurement
conditions ($C_{OA}$~20-600 µg m$^{-3}$) the following median mass yields ranges were found: 7-20% for furans, 10-25% for SAH, 14-32% for PAH, 9-24% for OxyAH, 23-46% for $OVOC_{c\geq6}$ and 6-18% for $OVOC_{c<6}$. Mass yields were also calculated for the two separate datasets and reported in Fig. S6 in order to assess differences driven by specific combustion regimes. Considering ambient relevant conditions of $C_{OA}$~50 µg m$^{-3}$ we note a generally good agreement between the two datasets except for OxyAH which exhibit higher median yields (representing smoldering phase) of ~25% in Set1 and only ~10% in
Set2 (representing flaming phase). Set2 on the other hand exhibited higher median yields for the $OVOC_{c\geq6}$ family (~20% compared to ~16 % for Set1).

The effect of temperature and OH exposure on OA concentrations and yields are shown in Fig. 10 for different primary OM loads (total primary gaseous and particulate $OM_P$ + NTVOCs) of 6, 60 and 600 µg m$^{-3}$. The range of temperatures investigated




varies between 255 and 315 K. We find a general increase in total OA concentration with increasing OH exposure, decreasing experimental temperature, and higher initial loads, as expected. The average increase in OA concentration is 0.001, 0.03 and 0.6 $\mu g\ m^{-3}\ K^{-1}$ for 6, 60 and 600 $\mu g\ m^{-3}$, respectively. Concerning SOA yields the temperature effect is also a function of OH exposure and aerosol load; SOA yields increase by 0.0001, 0.0006 and 0.002 $g\ g^{-1}\ K^{-1}$ on average for 6, 60 and 600 $\mu g\ m^{-3}$,

respectively, with a higher effect predicted at lower temperature. We note overall an average yield increase by a factor of 3-4 for a 10-fold increase in the primary OM loads at the highest OH exposure considered ($8\times10^{7}$ molec $cm^{-3}$ h). Set2 exhibits higher yields because of lower contributions from the $OVOC_{c<6}$ family, which does not produce significant amounts of SOA. SOA yields increase with increasing $C_{OA}$ due to additional partitioning but also due to changes in the chemical composition and volatility of SOA species since they age differently with different experimental temperature and concentrations.

Compounds with different oxygen to carbon ratios lead to different functionality, polarity and vapor pressure upon aging. Moreover, different temperatures result in different evaporation enthalpies that influence consequently the compounds' volatility and lifetime. The modeled SOA $\Delta H_{vap}$ for each family of precursors results to be 17.5 kJ $mol^{-1}$ after GA calculation. This value is within the ranges of values reported in literature, where values between 11 and 44 kJ $mol^{-1}$ were reported for biogenic and anthropogenic SOA precursors depending upon the reactant hydrocarbon mixture and $NO_x$ concentration. For

SOA formed both from α-pinene or toluene, a negative correlation between $\Delta H_{vap}$ and $NO_x$ concentration was observed (Offenberg et al., 2006).

The modelled fractional contributions of the six different precursor classes to SOA are shown in Fig. 11 for Set1 and Set2. The most dominant contribution is from the OxyAH family apart from Set1 at high OH exposure where the contributions from precursors with higher volatility (furans and $OVOC_{c<6}$) are more strongly temperature-dependent. In detail the $OVOC_{c<6}$ family

exhibits higher contribution with higher initial load and higher OH exposure. The SAH and PAH families have  relevant contributions for Set2 because these compounds are strongly emitted during the flaming phase. Since the SAH family consists of compounds that are less reactive than other families, they become relevant just at high OH exposure, while the compounds in the PAH family react faster and show a decreasing contribution with increasing OH exposure.

# 5. Conclusions

We performed box model simulations, based on the volatility basis set (VBS) approach, of residential wood combustion smog chamber experiments conducted at different temperatures, different combustion conditions and using different residential stoves. Primary emissions of SOA precursor compounds (VOCs) and organic aerosol (OA) as well as their evolution during aging in the smog chamber were simultaneously monitored by a PTR-ToF-MS and an HR-ToF-AMS, respectively. This enabled the identification of the nature of SOA precursors lumped into different classes according to their chemical

composition.

The knowledge about the nature of SOA precursors was used to better constrain model parameters, in the oxidation products production rates and elemental composition. Using the measured OA mass, we were able to determine the volatility





distributions and $\Delta H_{vap}$ for the products formed from the oxidation of the dominant precursor compound classes. We estimated the contributions of different compound classes to SOA and evaluate how the variability in the emission composition under a wide range of conditions would influence the SOA yield predictions. Investigation of different experimental temperatures allowed the evaluation of the model evaporation enthalpies which have a decisive influence on the volatility of the emissions

and hence their atmospheric lifetimes. Upon aging, compounds with lower atomic O:C ratios are converted through the oxidation pathway to products with higher functionality, higher polarity and lower vapor pressure. As a result, a part of these products (re-)condense to the particle phase with partial pressures determined by their volatility, ambient temperature and concentration of the particulate organic mixture. While the degree of oxygenation increases during aging, organic species may also fragment into more volatile compounds, being eventually converted into $CO_2$. Understanding the balance between

oxygenation and fragmentation, their effect on volatility of emissions and timescale of these processes is essential to predict the evolution of the OA concentration.

Overall we developed a framework useful to constrain complex emissions and suitable for sophisticated mass spectrometry analysis with the novelty and ability of identifying the contributions of different classes of VOC precursors to SOA formation. The main focus of the study included the investigation of smoldering versus flaming emissions, resulting in predominant

contributions of different classes of compounds according to the combustion phase investigated. Smoldering phase emissions were dominated by the $OVOC_{c<6}$ compound family while the flaming phase exhibited higher contributions by the SAH and PAH families. In general for both phases studied, higher contributions to SOA formation was found for cresol and phenol species and chemically similar compounds. These species were therefore predicted to be important markers to be monitored in air pollution studies in order to estimate the SOA forming potential from real emissions.

**Appendix A**

In this appendix, we present the derivation of the thermodynamic equations used in the model. We considered the species in the gas and the particle phase to be permanently at equilibrium, as condensation is expected to be faster than oxidation (time scales for oxidized vapours condensation < 1 minute assuming no particle phase diffusion limitations, Bertrand et al., 2018b). Accordingly, the relation between the aerosol and gas phase activities satisfies the following expression:

$$x_{i,j}|_g = \gamma_{i,j}\chi_{i,j}x_{i,j}^0 \tag{A1}$$

Here, $x_{i,j}|_g$ denotes the gas phase molar concentration of a surrogate in a volatility bin $i$ formed from a precursor class $j$. The product $\gamma_{i,j}\chi_{i,j}$ represents the activity of the same surrogate in the particle phase, where $\gamma_{i,j}$ and $\chi_{i,j}$ are the activity coefficient and the fraction of the surrogate in the particle phase, respectively. $x_{i,j}^0$ is the equilibrium molar saturation concentration of the

pure surrogate, related to its equilibrium vapor pressure, $p_{i,j}^0$, according to Eq. (A2).



$$x_{i,j}^0 = \frac{p_{i,j}^0}{RT} \tag{A2}$$

Here, $R$ and $T$ are the ideal gas constant and the temperature, respectively. The effective molar saturation concentration ($x_{i,j}^*$) which takes into account the influence of non-ideal mixing on the compounds' activity, can be defined as the product of $\gamma_{i,j}$ and $x_{i,j}^0$:

$$x_{i,j}^* = \gamma_{i,j} x_{i,j}^0 \tag{A3}$$

Similar to $p_{i,j}^0$, $x_{i,j}^*$ can be written as a function of temperature, according to the Clausius-Clapeyron relationship based on Eq. (A4).

$$\ln\left(\frac{x_{i,j}^*(T)}{x_{i,j}^*(T_{ref})}\right) = -\frac{\Delta H_{vap\,i,j}}{R}\left(\frac{1}{T} - \frac{1}{T_{ref}}\right) \tag{A4}$$

Here, $T$ and $T_{ref}$ are the experimental and reference $(T_{ref} = 298K)$ temperatures, respectively. $\Delta H_{vap\,i,j}$ is the effective enthalpy of evaporation. It includes the effects of temperature on (1) the pure compound vapor pressure ($p_{i,j}^0$), (2) the compounds' mixing properties in the condensed phase, i.e. $\gamma_{i,j}$ and (3) the radical chemistry reaction rate constants and branching ratios in the gas-phase (Stolzenburg et al., 2018).

$x_{i,j}^*$ is related to the Donahue effective mass saturation concentration (Donahue et al., 2012), $C_{i,j}^*$, which is the inverse of the Pankow equilibrium constant (Pankow, 1987), through the compounds' molecular weight, $MW_{i,j}$, as indicated in Eq. (A5).

$$C_{i,j}^* = MW_{i,j} x_{i,j}^* \tag{A5}$$

To constrain the model to the measurements, it is of convenience to rearrange Eq. (A1) and (A3) as a function of the surrogate total concentration, $x_{i,j}|_{g+p} = x_{i,j}|_g + x_{i,j}|_p$ and the total molar concentration of species in the particle phase ($x_{OA}$) which yields the following expression:

$$x_{i,j}|_p = \left(1 + \frac{x_{i,j}^*}{x_{OA}}\right)^{-1} x_{i,j}|_{g+p} \tag{A6}$$

The parameters in Eq. (A6) were equated as follows:

- The modelled molar concentration of the particulate organic phase was expressed as the sum of the concentration of all surrogates in the particle phase Eq. (A7).



$$x_{OA} = \sum_j^n \sum_i^m x_{i,j}|_p + x_i^{OM_p}|_p \tag{A7}$$

Here, $m$ and $n$ are the total number of volatility bins and precursors chemical classes, respectively. $x_i^{OM_p}|_p$ is the particle phase molar concentration of primary organic matter in volatility bin $i$. The $x_i^{OM_p}|_p$ is calculated based on $x_i^{OM_p}|_{g+p}$ in both phases, following a similar computation as for SOA (Eq. A6). $x_i^{OM_p}|_{g+p}$ was inferred from the measured POA concentrations injected into the chamber at the beginning of the experiment and using the volatility distribution function of wood combustion emissions in May et al. (2013).

- $x_{i,j}|_{g+p}$ was derived from the precursor oxidation rates measured by the PTR-ToF-MS. The change in the total concentration of oxidation products from a precursor class $j$ in both gas and particle phases, $x_j|_{g+p}$, was expressed as follows:

$$\frac{dx_j|_{g+p}}{dt} = k_{OH} * [OH] * [VOC_j] + k_{other} * [VOC_j] - k_{dil} * x_j|_{g+p} \tag{A8}$$

Here, $[VOC_j]$ is the molar concentration of total VOC precursors in class $j$. For the definition of the other parameters, the reader is referred to Eq. (1). $x_j|_{g+p}$ is calculated by numerically integrating Eq. (A8). $x_j|_{g+p}$ is related to the concentration of the different surrogates from a precursor class $j$ in different volatility bins, $x_{i,j}|_{g+p}$ (Eq. A6), through their yields, $\Upsilon_{i,j}$:

$$x_{i,j}|_{g+p} = \Upsilon_{i,j} x_j|_{g+p} \tag{A9}$$

These yields, which represent the surrogate volatility distributions were determined by the model.

**Authors contribution**

Main author: GS. Conceptualization: IEH, JGS, NM, ASHP. Experimental work: GS, AB, EB, SP. Formal analysis: GS, IEH, AB, EB. Model development and output analysis: JJ, IEH, SA. Supervision: JGS, IEH, ASHP, UB.





## Acknowledgments

This study was supported by the Swiss National Science Foundation (project WOOSHI (200021L_140590) and SNSF Starting Grant IPR-SHOP (BSSGI0_155846)), the European Community's Seventh Framework Programme (FP7/2007-2013) under grant agreement no. 290605 (PSI-FELLOW), the European Union's Horizon 2020 research and innovation programme through the EUROCHAMP-2020 Infrastructure Activity under grant agreement no. 730997, and the Competence Centers Environment and Sustainability (CCES) and Energy and Mobility (CCEM) (project OPTIWARES). AMU acknowledges support from the French Environment and Energy Management Agency (ADEME) under the grant 1562C0019 (VULCAIN project) and the Provence-Alpes-Côte d'Azur (PACA) region. We appreciate the availability of the VBS framework in CAMx and support of RAMBOLL.

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

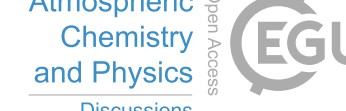



**Table 1. Experimental parameters for the 14 smog chamber experiments used in this study, including smog chamber temperature, stove type, modified combustion efficiency (MCE), and the intial concentrations of POA and VOCs.**

| Expt. | dataset | reference | date | experimental temperature (°C) | stove type | MCE | POA (μg m$^{-3}$) | VOCs (μg m$^{-3}$) |
|---|---|---|---|---|---|---|---|---|
| 1 | | Bertrand et al. 2017 | 29.10.2015 | 2 | stove 1 | 0.85 | 126 | 4039 |
| 2 | | Bertrand et al. 2017 | 30.10.2015 | 2 | stove 1 | 0.84 | 179 | 7862 |
| 3 | | Bertrand et al. 2017 | 04.11.2015 | 2 | stove 1 | 0.83 | 73 | 3694 |
| 4 | Set1 | Bertrand et al. 2017 | 05.11.2015 | 2 | stove 1 | 0.91 | 10 | 948 |
| 5 | | Bertrand et al. 2017 | 06.11.2015 | 2 | stove2 | 0.80 | 42 | 1839 |
| 6 | | Bertrand et al. 2017 | 07.11.2015 | 2 | stove2 | 0.87 | 35 | 2007 |
| 7 | | Bertrand et al. 2017 | 09.11.2015 | 2 | stove2 | 0.82 | 44 | 3379 |
| 8 | | Bruns et al. 2016, Ciarelli et al. 2017 | 02.04.2014 | -10 | stove 3 | 0.97 | 9 | 301 |
| 9 | | Bruns et al. 2016, Ciarelli et al. 2017 | 17.03.2014 | -10 | stove 3 | n.a. | 12 | 1024 |
| 10 | | Bruns et al. 2016 | 25.03.2014 | 15 | stove 3 | 0.97 | 22 | 526 |
| 11 | Set2 | Bruns et al. 2016 | 27.03.2014 | 15 | stove 3 | 0.97 | 15 | 645 |
| 12 | | Bruns et al. 2016 | 28.03.2014 | 15 | stove 3 | 0.97 | 17 | 1368 |
| 13 | | Bruns et al. 2016 | 29.03.2014 | 15 | stove 3 | 0.97 | 18 | 1096 |
| 14 | | Bruns et al. 2016 | 30.03.2014 | 15 | stove 3 | 0.97 | 18 | 910 |

5   n.a.: not available.

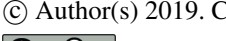



**Table 2. Reported average reaction rate constants ($10^{-11}$ cm$^3$ molec$^{-1}$ s$^{-1}$) towards OH per family at the beginning of each experiment, including average reactivity (AVG) and standard deviation (STDEV).**

| Expt. | Furans | SAH | PAH | OxyAH | OVOC$_{c \geq 6}$ | OVOC$_{c < 6}$ |
|---|---|---|---|---|---|---|
| 1 | 2.96 | 1.90 | 2.90 | 1.94 | 0.66 | 1.16 |
| 2 | 2.97 | 1.86 | 2.99 | 2.05 | 0.70 | 1.25 |
| 3 | 3.04 | 1.88 | 2.95 | 2.07 | 0.66 | 1.19 |
| 4 | 2.79 | 2.02 | 2.84 | 2.38 | 0.62 | 1.16 |
| 5 | 3.04 | 2.10 | 2.73 | 2.57 | 0.60 | 1.12 |
| 6 | 3.04 | 1.88 | 2.83 | 2.09 | 0.59 | 1.11 |
| 7 | 2.92 | 1.93 | 2.80 | 2.05 | 0.67 | 1.21 |
| 8 | 3.09 | 2.28 | 3.36 | 3.74 | 0.88 | 1.40 |
| 9 | 3.65 | 2.37 | 3.76 | 5.58 | 0.69 | 1.48 |
| 10 | 3.15 | 2.34 | 3.27 | 4.41 | 0.59 | 1.44 |
| 11 | 3.11 | 2.34 | 3.07 | 4.04 | 0.60 | 1.57 |
| 12 | 3.04 | 2.23 | 2.50 | 2.32 | 0.75 | 1.39 |
| 13 | 3.02 | 2.33 | 3.48 | 2.98 | 0.72 | 1.41 |
| 14 | 2.96 | 2.36 | 3.58 | 4.40 | 0.77 | 1.50 |
| AVG | 3.06 | 2.13 | 3.08 | 3.04 | 0.68 | 1.31 |
| STDEV | 0.19 | 0.21 | 0.36 | 1.17 | 0.08 | 0.16 |





**Figure 1. Schematic of the modelling framework. The box model simulates the formation of SOA from each precursor class *j* in volatility bins *i*. The best solution of the initialized input parameters volatility distribution (described as mean value of log$C^*$ for each precursor class μ$_j$ and standard deviation σ) and enthalpy of vaporization ($\Delta H_{vap}$) parameters are optimized by a genetic algorithm, using minimum mean bias and root mean square error (RMSE) between modelled and measured OA concentration as the fitness function. Green boxes represent measured data from the chamber experiments.**





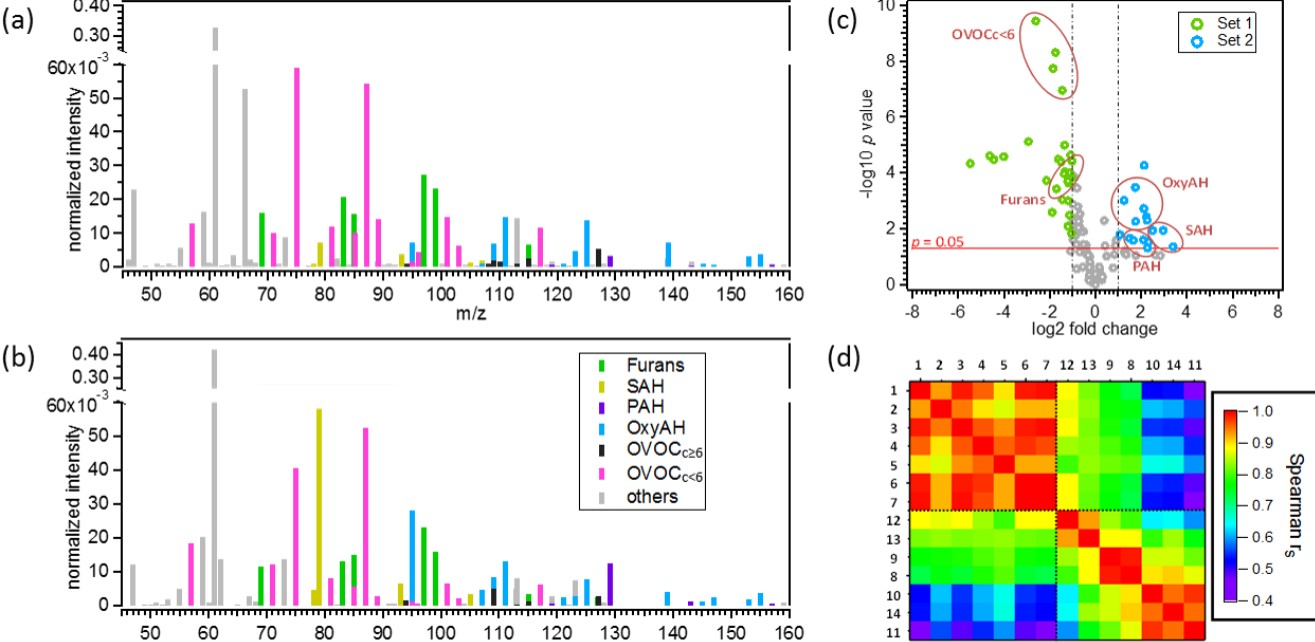

**Figure 2. Comparison of the VOC emissions between the two sets of experiments. Panels (a) and (b) display average primary VOCs mass spectra from Set1 (representing the smoldering phase) and Set2 (representing mainly the flaming phase), respectively. Spectra are normalized to the initial total VOC concentration in μg m⁻³. Compounds are color coded by chemical classes. c) *p*-value versus fold change comparing the fingerprints of primary VOCs between the two sets of experiments. The fold change was calculated as the ratio of the intensities of each ion normalized to the total signal, between Set2 and Set1 averaged across experiments. Data points above *p* = 0.05 have significantly different contributions to the total VOCs between the two sets of experiments. Blue colored data points on the right hand side designate compounds enriched in the emissions from Set2 while green colored data points on the left hand side designate compounds enriched in the emissions from Set1. d) Spearman correlation matrix for the primary VOCs mass spectra between all experiments highlighting the variability in the composition of the primary emissions. Each experiment is identified by an index (see Table 1) and the experiments from Set2 were reordered according to similarity with Set1.**



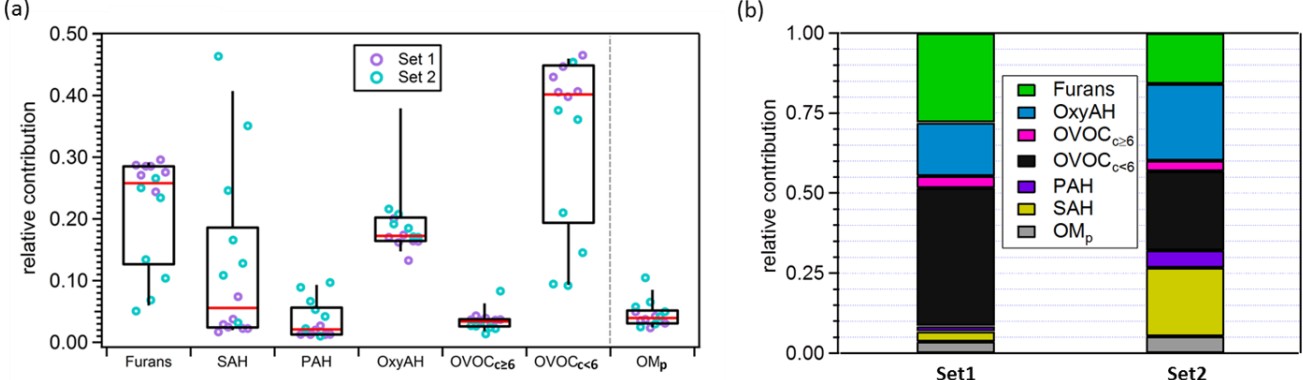

**Figure 3. a) Box plot for the relative contributions to the total primary emissions of the different precursor classes and of OM$_p$ averaged between experiments. OM$_p$ is the total semi-volatile and low-volatile organic matter calculated by means of the VBS model**
**assuming the volatility distribution from May et al. (2013) and using as input the measured organic aerosol (OA) mass. The top and bottom whiskers represent the 90$^{th}$ and 10$^{th}$ percentiles, respectively, while the top, middle and bottom lines of the boxes show the 75$^{th}$, 50$^{th}$ and 25$^{th}$ percentiles, respectively. The circles represent each single experiment from the two datasets investigated. b) Average contributions of different precursor families and OM$_p$ for Set1 and Set2.**



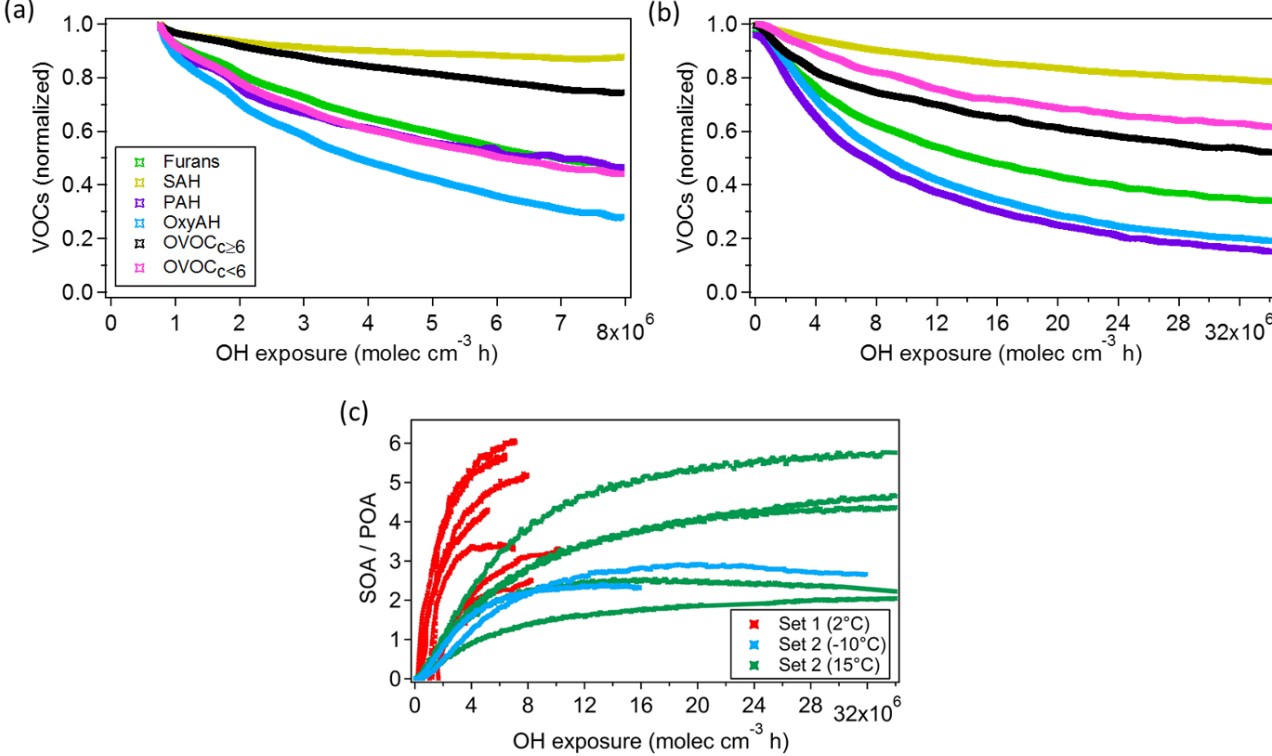

**Figure 4.** Average consumption of SOA precursor classes against average OH exposure for Set1 (a) and Set2 (b). The observed decay of SOA precursor families (VOCs) as described in Eq. (1) is due to both oxidation processes and dilution in the chamber. c) SOA to POA ratio for each experiment against average OH exposure colored according to experimental temperatures (2°C, -10°C, 15°C).





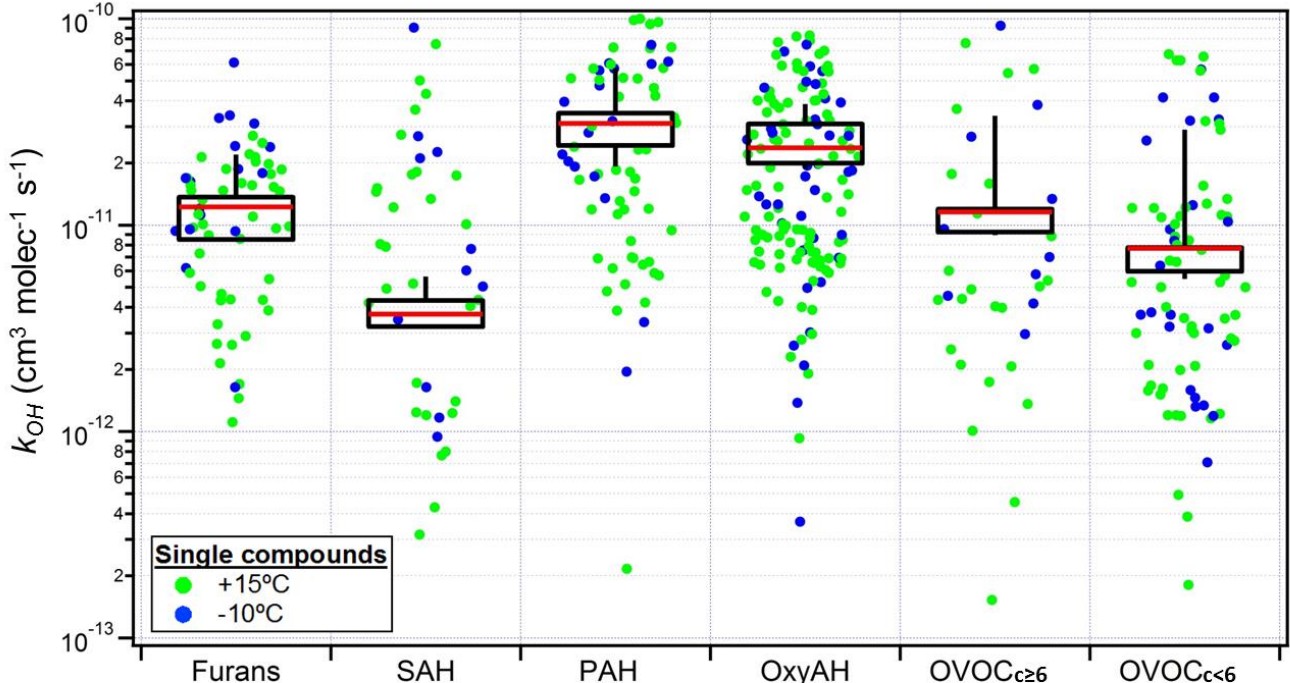

**Figure 5.** Box plot of mass weighted average OH reaction rate constants ($k_{OH}$) determined for each precursor class from Bruns et al. (2016) for Set2 only (see Table S3). The individual $k_{OH}$ values for all compounds are also shown for all experiments, color coded according to the experimental temperatures. The top and bottom whiskers represent the 90[th] and 10[th] percentiles, while the top, middle and bottom lines of the boxes show the 75[th], 50[th], and 25[th] percentiles, respectively.

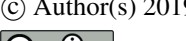



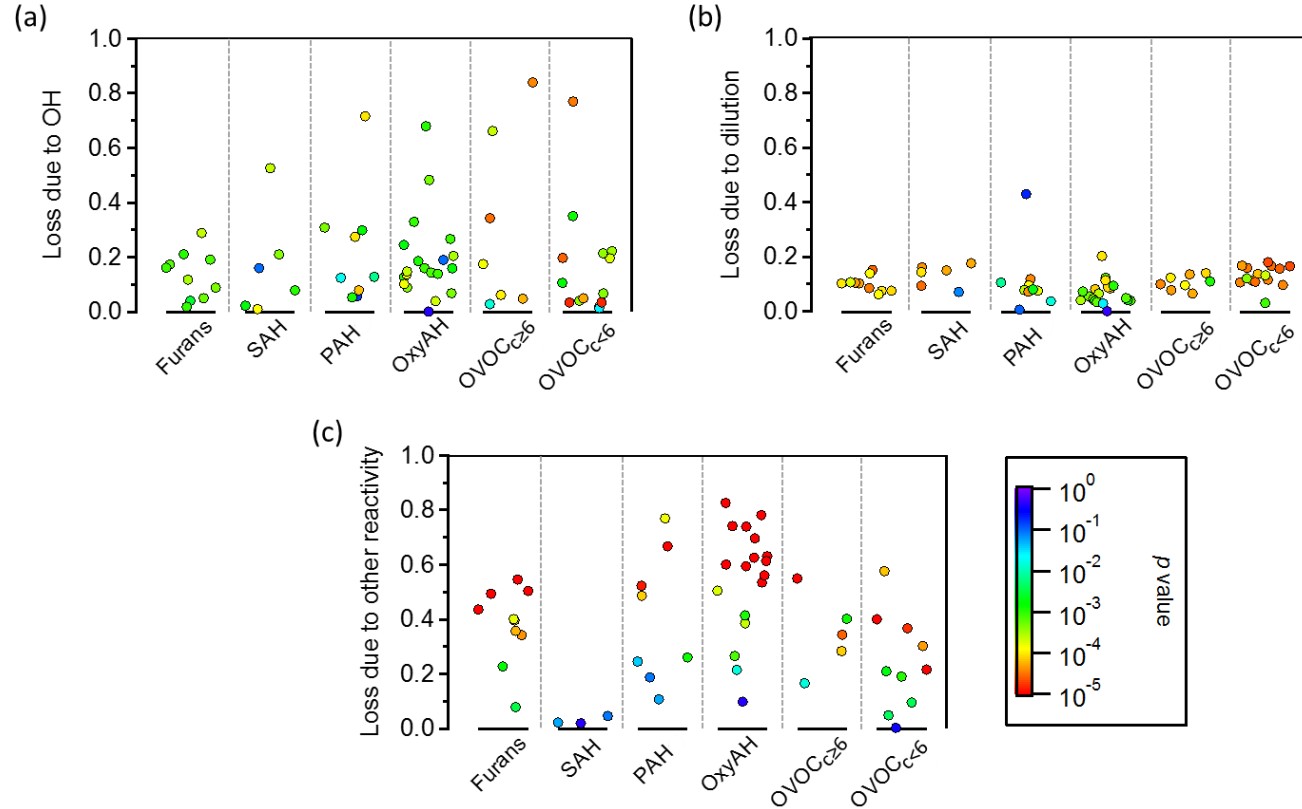

**Figure 6. Fraction of consumed precursor compounds for Set1 by OH oxidation (a), dilution (b) and other reactivity (c) at the end of the experiments. Each point corresponds to a single compound averaged among experiments and normalized to the initial concentration. The color legend represents the statistically significant deviation from zero reactivity with the investigated reactant (*p* value).**





**Figure 7. a) Comparison between the sum of simulated organic aerosol (OA) concentrations from different precursor families and the measured OA concentration for each smog chamber experiment. Set1 (Expt. 1-7, 2°C) and Set2 (Expt. 8-9, -10°C and Expt. 10-14, 15°C). b) Probability distribution of mean bias weighted by the average measured OA concentration. The resulting mean bias is -7.2 μg m⁻³ (~15%) and the root-mean-square error (RMSE) is 37.4 μg m⁻³. The model tend to underestimate the measured OA for Set1 with a mean bias of 7% while it tends to overestimate the measured OA for Set2 with a mean bias of 8%. OH exposure is presented in convenient way for visualization purpose.**





**Figure 8. a) Comparison between modelled and measured O:C ratio for each experiment. Set1 (Expt. 1-7, 2°C) and Set2 (Expt. 8-9, -10°C and Expt. 10-14, 15°C). b) Probability distribution of the relative bias (normalized by the averaged measured O:C ratios). The resulting mean relative bias is 0.006 and the root-mean-square error (RMSE) is 0.06. OH exposure is presented in convenient way for visualization purpose.**





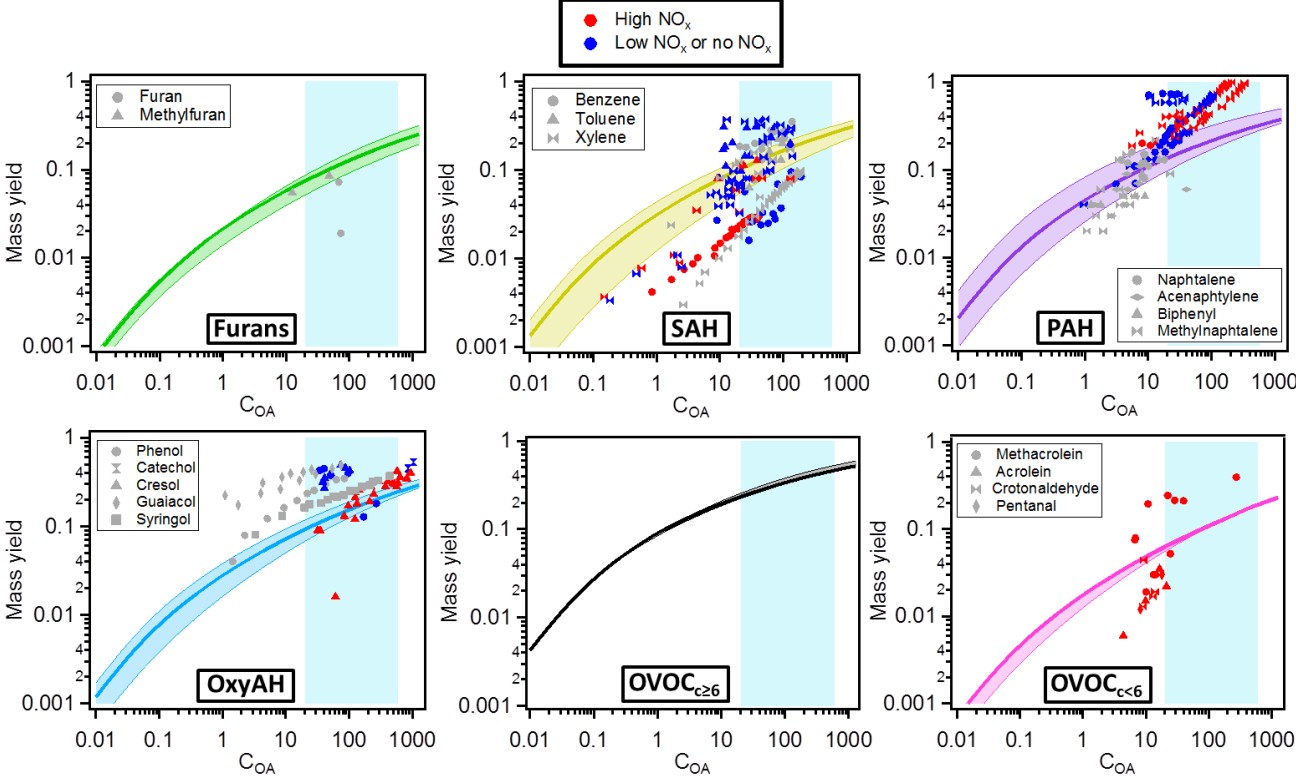

**Figure 9.** Mass yields for each class of compounds. The solid lines represent the median values while the lower and upper limits are the 25[th] percentile and 75[th] percentiles, respectively. The different markers in each plot are yields published in the literature for different single compounds (see Table S2). The color code denotes different NO$_x$ regimes (red denoting high NO$_x$, blue low or no NO$_x$ and grey not specified NO$_x$ regimes). The shaded background represents the range of our experiments (20-600 µg m$^{-3}$), outside this shaded area yields are extrapolated from the model.

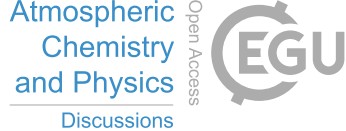



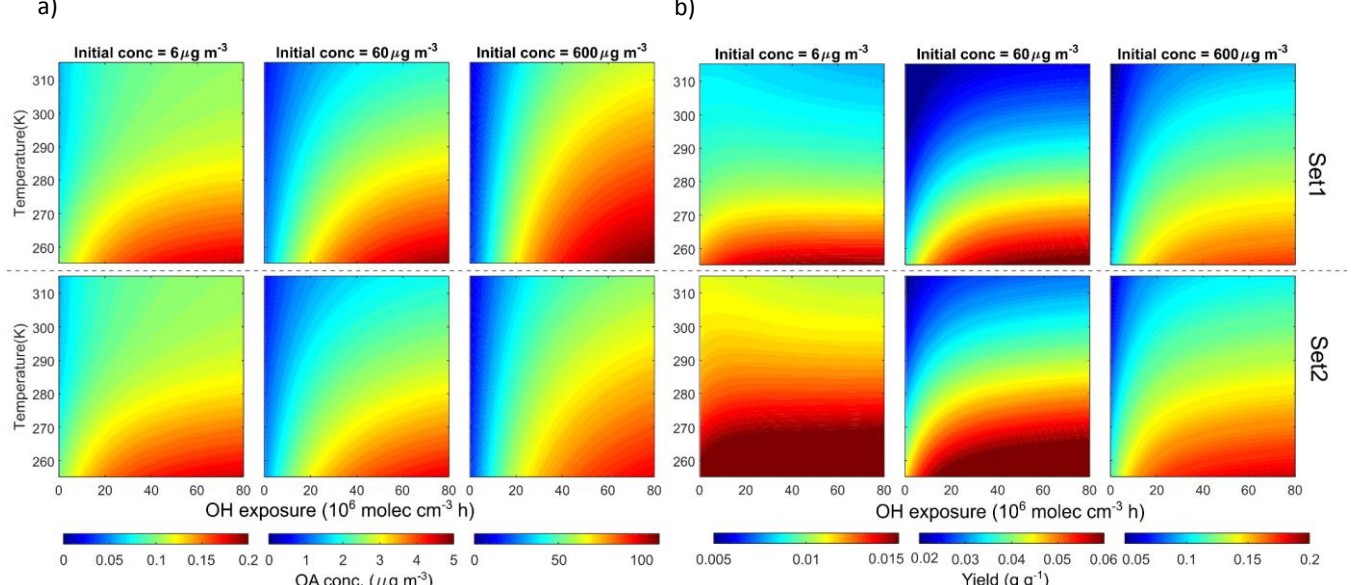

**Figure 10. Modelled OA concentrations (a) and yields (b) under different temperature, OH exposure and initial OA concentrations (6, 60 and 600 μg m⁻³). Upper and lower panels are based on Set1 and Set2, respectively. Temperature is provided in Kelvin (K) to avoid confusion with the experimental data in Celsius (°C).**



**Figure 11. Fractional contributions of the six precursor classes to SOA formation under different temperature, OH exposure, and initial OA concentrations. Left and right panels are based on Set1 and Set2, respectively. Temperature is provided in Kelvin (K) to**
5 **avoid confusion with the experimental data in Celsius (°C).**