# Peer review of "Secondary organic aerosol formation from smoldering and flaming combustion of biomass: a box model parametrization based on volatility basis set"

_Atmospheric Chemistry and Physics, 2018_

## Referee Comment (RC1) · Anonymous Referee #1 · 29 Jan 2019

**Review of ACP 2018-1308**

**Stefenelli et al.**

**Summmary:**

The authors burned wood in residential stoves under two burning conditions (flaming, smoldering) and with three different types of stoves. Emissions were sampled into a smog chamber and aged via OH-radical initiated oxidation. Several aging temperatures and VOC concentrations were explored. Resulting particles were measured with AMS and gas phase species with PTR-ToF. Gas-phase precursor species were sorted by chemical structure. SOA yields were measured. A box model was used to determine the volatility distribution of oxidation products. This was accomplished by iteratively adjusting the estimated concentrations of product species in various volatility bins, and the average enthalpy of vaporization, so that measured SOA concentrations best matched modeled concentrations. The relative contributions of different precursor VOC classes to SOA yield were determined using a genetic algorithm. The major results of the study are (1) smoldering and flaming conditions emit different primary VOCs, (2) cresols and phenols are important SOA precursors, (3) SOA yields increase at lower temperatures. However, the identification of which chemical classes of VOC are major SOA precursors was not consistent between different sets of experiments.

Overall I think that this work takes a promising approach to a difficult and important problem (SOA formation from biomass burning). The data set is interesting and the experimental techniques appropriate. However, there are some issues with disorganization in describing the analytical approach, the model implementation, and comparison of the effects of different variables (e.g. chemical composition, temperature, OH exposure) that make it difficult to follow the major conclusions and find information that could be useful for air quality modeling parameterization.

Maj**or comments:**

The analysis in this paper rests on the idea that there should be systematic differences in SOA yield between the six selected classes of SOA precursors. These classes are defined by structure: furans, PAH, unoxygenated aromatics, oxygenated aromatics, and two catch-all categories of other compounds with more and fewer than 6 carbon atoms. However, it is never clearly established why these six classes were chosen and so defined. Why not sort compounds by functionality (e.g. ketones, acids, diols, etc), O:C ratio, molecular weight, flaming vs. smoldering source, or some other characteristic? Why not handle each species separately? To summarize, the authors need to establish (1) why the lumping was necessary, and (2) why this particular differentiation of chemical characteristics should be expected to explain variability in SOA formation. Secondly, it appears that the individual species included in each chemical class were not the same for each experiment. This is a significant weakness of the analysis. The authors should strongly consider a more consistent and chemical meaningful approach to grouping compounds.

An additional point of concern is that multiple variables changed at same time: e.g. set 1 and set 2 experiments have different flaming/smoldering condition, different smog chamber temperatures, different OH exposure, and different VOC concentration. Is there a way to deconvolve these effects? The SOA yield of each experiment should best be compared at the same OH exposure.

Two papers using this dataset have been published previously (Bruns et al. 2016, 2017). In the introduction, the authors should more directly state the relationship of this work to the two previous papers and what new analysis is added.

The abstract could use some organizational editing, to clarify the major aims of the study, the analytical methods used to interpret the data, and the major results.

**Specific comments:**

It is not clearly described how different "flaming" and "smoldering" conditions were achieved in the two sets of experiments, and how the relative levels of flaming and smoldering were determined. The effect of temperature is not noted in the abstract.

Abstract: From the abstract it appears that one of the major conclusions is that flaming and smoldering conditions produce a different mix of VOCs, but this is not discussed in the paper.

Page 4 Line 1: These values don't quite make sense to me; how is it possible that the contributions consistently add to more than 100%? Is this mass yield or carbon yield?

Page 4 Line 9: Which parameters?

Page 5 line 11: If relative humidity was constant, than actual water vapor mixing ratios were quite different between the three temperature conditions. Is this expected to have an effect? Why was this particular humidity condition chosen?

Page 5: What NOx levels and NOx:VOC ratios were present?

Page 6 line 15: AMS collection efficiencies can be substantially less than 1, especially for very low volatility and highly oxidized particles. A collection efficiency correction could change the conclusions of this work. Can you support the assumption of CE=1?

Page 8 line 27: Was photolysis of VOCs considered?

Page 9 Line 7: How were these 263 ions selected?

Eq. 4 and 5: Could the measured OA be simultaneously corrected for wall loss and dilution by dividing by the measured BC signal over time, normalized to 1 at t=0?

Page 11 Lines 30-Page 12 line 9: This needs to be better supported and more detailed. Each of the six main precursor classes was further subdivided into six volatility bins, and the average chemical properties were determined for each bin – is this correct? Was there a large range in #C, #H, #O within each bin? It is not entirely clear to me how the volatility bins were determined: was the volatility of each individual species determined, and species were then grouped into bins? Or were species first grouped using some other method, then the volatility of each group was determined?

Page 13 lines 1-7: Can you explain more clearly what mutation, crossover, and selection mean for the implementation of the genetic algorithm in this particular application? How does the algorithm actually identify the optimal set of parameters?

Page 14 lines 21-23: It should be stated more clearly before this point that the individual species contributing to each of the 6 chemical classes were not consistent between different experiments. The

inconsistency in chemical composition makes it extremely difficult to compare between different experiments or to draw conclusions about the SOA yield from different chemical classes.

Page 15 lines 4-6: Two groups of compounds could have substantially different distribution of OH reactivities and SOA formation potential, but similar average OH reactivity. The conclusion here is not well supported.

**Technical corrections:**

Abstract 8-10: Sentence is difficult to understand, please rephrase.

Page 5 line 49: should be sodium nitrite, NaNO2

Eq. 1 Brackets for the sum term are missing

---

## Referee Comment (RC2) · Anonymous Referee #2 · 13 Mar 2019

**Comment on "Secondary organic aerosol formation from smoldering and flaming combustion of biomass: a box model parametrization based on volatility basis set" by Giulia Stefenelli et al. (2019)**

**Summary/recommendation:**

This paper is an interesting study on secondary organic aerosol formation from biomass burning. The authors conducted 14 experiments under two burning conditions (flaming, smoldering and flaming) and with different types of stove. Emissions from the burned stoves were sampled and were aged via OH-oxidation reactions to investigate the secondary organic aerosol yields and the chemical properties of the oxidation products from smoldering and flaming. A box model and a genetic algorithm approach were used to quantify the contribution of the VOC oxidation products to SOA yield and to better explain the SOA formation process.

However, many portions of the paper were made difficult to follow due to missing details. I recommend that this study be published but with minor revisions. I request that the authors consider the following points as they revise this manuscript:

**General comments:**

1/ As the title indicates, this paper should focus mainly on SOA formation from smoldering and flaming combustion of biomass. However, few information is given in that issue. The authors should better describe the burning conditions of smoldering and flaming by adding more details especially on how the authors reproduce experimentally the flaming and smoldering combustion.

2/ According to Majdi et al. (2019), Koo et al. (2014), Konovalov et al. (2015) and Ciarelli et al. (2017), Intermediate and Semi Volatile Organic Compounds (I-SVOCs) are considered as one of the most important SOA precursors emitted by biomass burning. Why did the authors focus only on SOA from VOCs ?

**Specific comments:**

1/ Page 4, lines 31-32: Why did the authors choose these experiments? How could the authors study the effect of smoldering/flaming combustion of biomass when the other experimental parameters (OH exposure, temperature of the smog chamber, stove.... ) change at the same time ?

2/ Page 5, line 27: Flaming combustion occurs at high temperature. Can the authors give an order of magnitude of this high temperature ?

3/ Page 5, line 31: Why did the authors choose to combine flaming and smoldering emissions in set 1?

4/ Page 7, line 7: What did the authors mean by "other processes"?

5/ Page 9, line 6: Can the authors give more information about this common set of 263 ions used to identify the most important SOA precursors?

6/ Page 11, line 22-23: Why did the authors choose to set the number of volatility bins to 6 ? How was the volatility $C^*$ determined ? Did the authors measure the volatility of each species ?

7/ Page 11: More clarification is needed in section 3.3.2. Did the given chemical properties of surrogates represent the average of the compounds that are classified in a determined volatility bin ?

8/ Page 12, line 11: How did the authors assume this single ΔH value for all surrogates? Did the authors consider any assumptions to determine this value?

9/ Page 13, section 3.3.4 is very short. Can the authors clarify how did this algorithm work to identify the optimized parameters and what did the authors mean by the process of "natural selection"?

10/ Page 13, Figure 2:  The 'others' VOC emissions show the highest relative contribution mainly in set 1. What are these 'others' VOCs emissions?

11/ Page 14,  line 11:  "Relevant compounds in the latter class are benzene, toluene and xylene." According to Bruns et al. (2016) experimental SOA yields, benzene is the third principal contributor to SOA after phenol and naphthalene. Did the authors characterize these SAH compounds?

12/ Page 14, line 20: "The SOA/POA ranges between 2 and 6, similar to ratios observed in previous studies." Please add references.

13/ Page 14, lines 22-23: "Such inconstancy in behavior is either due to differences in the chemical composition within the same class." How did the authors defined these chemical compositions? And how can these compositions change within the same class? Can this be explained by differences in the burned stoves? How can the authors compare SOA contribution from the different classes if the chemical composition can change? Please  explain this further.

14/ Page 18, lines 17-18: "In general for both phases studied, higher contributions to SOA formation was found for cresol and phenol species and chemically similar compounds." This conclusion is not well supported and not discussed in the paper.  Please clarify or add a reference.

**Technical comments:**

1) Figure 7 caption: Please replace "the model tend to underestimate ... "by "the model tends to underestimate...".

2) Eq (1) page 7: Is there any missing bracket in the equation ? Please verify.

3) Figure 2 caption: Did Set1 represent only smoldering phase or smoldering and flaming as defined in page 5 line 31? Please correct.

**References:**

Bruns, E., El Haddad, I., Lowik, J., Kilic, D., Klein, F., Baltensperger, U., and Prévôt, A.: Identification of significant precursor gases of secondary organic aerosols from residential wood combustion, Sci. Rep., https://doi.org/10.1038/srep27881-2016, 2016.

Ciarelli, G., El Haddad, I., Bruns, E., Aksoyoglu, S., Möhler, O., Baltensperger, U., and Prévôt, A. S. H.: Constraining a hy- brid volatility basis-set model for aging of wood-burning emissions using smog chamber experiments: a box-model study based on the VBS scheme of the CAMx model

(v5.40), Geosci. Model Dev., 10, 2303-2320, https://doi.org/10.5194/gmd-10- 2303-2017, 2017.

Konovalov, I. B., Beekmann, M., Berezin, E. V., Petetin, H., Mielonen, T., Kuznetsova, I. N., and Andreae, M. O.: The role of semi-volatile organic compounds in the mesoscale evolution of biomass burning aerosol: a modeling case study of the 2010 mega-fire event in Russia, Atmos. Chem. Phys., 15, 13269-13297, https://doi.org/10.5194/acp-15-13269-2015, 2015.

Koo, B., Knipping, E., and Yarwood, G.: 1.5-Dimensional volatility basis set approach for modeling organic aerosol in CAMx and CMAQ, Atmos. Environ., 95, 158-164, https://doi.org/10.1016/j.atmosenv.2014.06.031, 2014.

Majdi, M. , Turquety, S.,  Sartelet, K.,  Legorgeu, C., Menut, L. and Kim, Y.: Impact of wildfires on particulate matter in the euro-mediterranean in 2007: sensitivity to the parameterization of emissions in air quality models. Atmos. Chem. Phys., 19, 785-812, https://doi.org/10.5194/acp-19- 785-2019, 2019.

---

## Author Comment (AC1) · 20 Jun 2019

**Response to reviewer's comments:**

**Title:** Secondary organic aerosol formation from smoldering and flaming combustion of biomass: a box model parametrization based on volatility basis set

**Journal:** Atmospheric Chemistry and Physics

**Manuscript ID:** acp-2018-1308

Dear Editor,

We thank the reviewers for their comments. Our detailed point-by-point responses to the reviewers' comments (in black regular typeset) are provided in blue regular typeset and the revised text (highlighted in the main text) is in *grey italic typeset*.

**Reviewer 1:**

**Summary:**

The authors burned wood in residential stoves under two burning conditions (flaming, smoldering) and with three different types of stoves. Emissions were sampled into a smog chamber and aged via OH-radical initiated oxidation. Several aging temperatures and VOC concentrations were explored. Resulting particles were measured with AMS and gas phase species with PTR-ToF. Gas-phase precursor species were sorted by chemical structure. SOA yields were measured. A box model was used to determine the volatility distribution of oxidation products. This was accomplished by iteratively adjusting the estimated concentrations of product species in various volatility bins, and the average enthalpy of vaporization, so that measured SOA concentrations best matched modeled concentrations. The relative contributions of different precursor VOC classes to SOA yield were determined using a genetic algorithm. The major results of the study are (1) smoldering and flaming conditions emit different primary VOCs, (2) cresols and phenols are important SOA precursors, (3) SOA yields increase at lower temperatures. However, the identification of which chemical classes of VOC are major SOA precursors was not consistent between different sets of experiments.

Overall I think that this work takes a promising approach to a difficult and important problem (SOA formation from biomass burning). The data set is interesting and the experimental techniques appropriate. However, there are some issues with disorganization in describing the analytical approach, the model implementation, and comparison of the effects of different variables (e.g. chemical composition, temperature, OH exposure) that make it difficult to follow the major conclusions and find information that could be useful for air quality modeling parameterization.

**Major comments:**

The analysis in this paper rests on the idea that there should be systematic differences in SOA yield between the six selected classes of SOA precursors. These classes are defined by structure: furans, PAH, unoxygenated aromatics, oxygenated aromatics, and two catch-all categories of other compounds with more and fewer than 6 carbon atoms. However, it is never clearly established why these six classes were chosen and so defined. Why not sort compounds by functionality (e.g. ketones, acids, diols, etc), O:C ratio, molecular weight, flaming vs. smoldering source, or some other characteristic? Why not handle each species separately? To summarize, the authors need to establish (1) why the lumping was

necessary, and (2) why this particular differentiation of chemical characteristics should be expected to explain variability in SOA formation. Secondly, it appears that the individual species included in each chemical class were not the same for each experiment. This is a significant weakness of the analysis. The authors should strongly consider a more consistent and chemical meaningful approach to grouping compounds.

The reviewer has raised two concerns in this comment. We have broken the comment into two specific points, addressing each in turn.

**Point 1: the rationale behind the lumping approach:**

As it can be seen from the relatively high uncertainties in the yields (up to factor of three, Figure 9), the size of our dataset does not allow us retrieving the volatility distribution for the single precursors, which would entail the determination of more than 80 free parameters. This is especially the case as the precursor time series, decaying with oxidation, are typically strongly correlated, which prevents resolving systematic differences between the yields of the different single precursors. Indeed, the current state-of-the-art VBS applications lump all SOA precursors in one single group (e.g. Ciarelli et al. 2017). Therefore, lumping is needed to decrease the model degree of freedom and currently, the model is solved for eight free parameters. The lumping approach is based on the two objectives of the study:

[revised manuscript text omitted]

**Point 2: Selected precursors in Set1 and Set2:**
The individual species included in each chemical class are the same for each experiment and for both the datasets. We are sorry for the misunderstanding and we add the following statement in the end of section 3.1.4.:
*Page 9 line 16: "The selected precursors in each class are the same for each experiment and each dataset."*

An additional point of concern is that multiple variables changed at same time: e.g. set 1 and set 2 experiments have different flaming/smoldering condition, different smog chamber temperatures, different OH exposure, and different VOC concentration. Is there a way to deconvolve these effects? The SOA yield of each experiment should best be compared at the same OH exposure.

Extracting class-specific yields essentially resides on the variability in the contribution of different classes to condensable species, between experiments (due to variability in the emissions) or within an experiment (due to differences in the precursors reaction rates). By nature, emissions are very complex and therefore hard to control and typically several parameters co-vary. It is important in fitting for the class specific yields in such a multivariate problem that the contribution of the different precursors to the condensable gases do not co-vary with other parameters (e.g. temperature, OH exposure), otherwise their respective effects cannot be distinguished and biases in the yields may arise. The main variability in the contribution of different classes to condensable species is due to variability in the emissions between Set1 and Set2. Most of the other parameters in Set1 and Set2 are consistent or have a significant overlap, allowing the retrieval of class specific yields and a direct comparison between the experiments. In the following, we compare Set1 and Set2 with respect to the different combustion and aging parameters:

➔ Combustion conditions: The combustion conditions between the two sets are comparable, in terms of the starting procedure and the type and the amount of wood burned. The main difference is related to the air input, which favored smoldering and flaming conditions for Set1 and Set2, respectively. This is directly reflected in the composition of the SOA precursors and such variability is key to deconvolve the class-specific yields. Despite this, the emission composition for Set2 is variable, with some experiments showing a similar composition as Set1 (see Fig. 2d), ensuring an overlap between Set1 and Set2 chemical composition when other parameters varied.

➔ OH exposure: The OH exposure is ~7 times higher for Set2 compared to Set1. This is because the OH production rate was similar between the two experiments, but its sink was significantly higher for Set1 due to the injection of higher emissions. However, we note that the OH exposure is not directly considered in the model. Instead, the integrated production of condensable species, inferred from the precursor decay, is used as inputs, which takes into account the differences in the OH exposures. Besides, we note that most of the SOA production occurs at OH exposures $< 8 \cdot 10^6$ molecules cm$^{-3}$ h, which is reached in both experiments.

➔ Emission loading: On average, higher concentrations were injected during Set1 compared to Set2. This was not intentional, but occurred as smoldering emissions rates were much higher and therefore it was more challenging to control the amounts injected. That said it is important that the concentrations span a wide range (here 1.5 orders of magnitude) to be able to constrain the amounts of oxidation products in the different volatility bins. As mentioned in the manuscript, the volatility distributions are well constrained in the saturation concentrations between 10-1000 µg m$^{-3}$, corresponding to the range of OA concentrations studied here. We finally note that the concentration ranges in Set1 and Set2 have a significant overlap (for Set1, SOA = 50-800 µg m$^{-3}$ vs. 20-140 µg m$^{-3}$), ensuring that the fitting is not biased due to difference in the loadings between the two sets.

➔ Aging temperature: conducting the experiments at different temperatures is essential for constraining the condensable species enthalpy of evaporation. Set1 experiments were conducted at 2°C, right in the middle of the two temperatures examined in Set2: -10°C and 15°C. Therefore, the temperature does not co-vary with the changes in the emission composition between Set1 and Set2 and the retrieval of the yields is not affected by the temperature.

Finally, we have assessed the sensitivity of the yield retrieval on the experiments included in the parameterization, through the bootstrap analysis. This influences the covariance between parameters and therefore this sensitivity analysis is a way to assess the biases resulting from such covariance. We present the resulting ranges of yield parameters from the sensitivity analysis.

Two papers using this dataset have been published previously (Bruns et al. 2016, 2017). In the introduction, the authors should more directly state the relationship of this work to the two previous papers and what new analysis is added.

Based on the reviewer comment, we have modified the introduction to clarify the links between the current study and Bruns et al. 2016 and 2017. The modified text reads as follows:

*"Bruns et al. (2016) investigated the SOA formation from residential log wood combustion from a single type of stove under stable flaming conditions only. They reported that T-SOA precursors included in models account for only 3 to 27% of the measured SOA whereas 84 to 116% was from NT-SOA precursors including in total 22 individual compounds and two lumped compound classes, mainly consisting of polycyclic aromatic hydrocarbons from incomplete combustion (e.g. naphthalene) and cellulose and lignin pyrolysis products (e.g. furans and phenols, respectively). The estimated SOA*

*concentrations were based on the literature SOA yields of single precursors, obtained from smog chamber experiments, and a good agreement was observed between predicted and measured SOA. However, the method suffers from two drawbacks. First, the dependence of the yields on the organic aerosol loading and temperature was not considered. Second, although the relative contributions of different precursors to SOA were estimated, thermodynamic parameters for chemical transport models (CTMs) were not determined. Based on the same experiments, the lumped concentrations of the 22 non-traditional volatile organic compounds and 2 compound classes were constrained in a box model (Ciarelli et al., 2017a). Improved parameters were retrieved describing the volatility distributions and the production rates of oxidation products from the overall mixture of precursors present in biomass smoke. While this method is well suited for CTMs (Pandis et al., 2013; Ciarelli et al., 2017b), it does not provide any information about the contributions of the different chemical classes to the aerosol."*

We have also explicitly defined the two sets of experiments in the method section:

*"Experiments from Bertrand et al., 2017, 2018a will be referred to as Set1 and experiments from Bruns et al., 2016 and Ciarelli et al., 2017 as Set2."*

The abstract could use some organizational editing, to clarify the major aims of the study, the analytical methods used to interpret the data, and the major results.

Based on the reviewer comment, we have reorganized the abstract and highlighted more clearly the major aims of the study. The modified abstract reads as follows:

*"Residential wood combustion remains one of the most important sources of primary organic aerosols (POA) and secondary organic aerosol (SOA) precursors during winter. The overwhelming majority of these precursors have not been traditionally considered in regional models, and only recently, lignin pyrolysis products and polycyclic aromatics were identified as the principal SOA precursors from flaming wood combustion. The SOA yields of these components in the complex matrix of biomass smoke remain unknown and may not be inferred from smog chamber data based on single compound systems. Here, we studied the aging of emissions from flaming and smoldering-dominated wood fires in three different residential stoves, across a wide range of aging temperatures (-10°C, 2°C and 15°C) and emission loads. Volatile organic compounds (VOCs) acting as SOA precursors were monitored by a proton transfer reaction time-of-flight mass spectrometer (PTR-TOF-MS), while the evolution of the aerosol properties during aging in the smog chamber was monitored by a high resolution time-of-flight aerosol mass spectrometer (HR-ToF-AMS). We developed a novel box model based on the volatility basis set (VBS) to determine the volatility distributions of the oxidation products from different precursor classes found in the emissions, grouped according to their emission pathways and SOA production rates. We show for the first time that SOA yields in complex emissions are consistent with those reported in literature from single compound systems. We identify the main SOA precursors in both flaming and smoldering wood combustion emissions at different temperatures. While single-ring and polycyclic aromatics are significant precursors in flaming emissions, furans generated from cellulose pyrolysis appear to be important for SOA production in the case of smoldering fires. This is especially the case at high loads and low temperatures, given the higher volatility of furan oxidation products predicted by the model. We show that the oxidation products of oxygenated aromatics from lignin pyrolysis are expected to dominate SOA formation, independent of the combustion or aging conditions, and therefore can be used as promising markers to trace aging of biomass smoke in the field. The model framework developed herein may be generalizable for other complex emissions sources, allowing determination of the contributions of different precursor classes to SOA, at a level of complexity suitable for implementation in regional air quality models."*

**Specific comments:**

It is not clearly described how different "flaming" and "smoldering" conditions were achieved in the two sets of experiments, and how the relative levels of flaming and smoldering were determined. The effect of temperature is not noted in the abstract.

Ward and Hardy (1991) define the flaming and smoldering conditions according to the modified combustion efficiency, MCE = $CO_2$/(CO+$CO_2$). Specifically, MCE > 0.9 is identified as flaming condition, while MCE < 0.85 is identified as smoldering condition. We have already reported these values in Table 1. Practically, we achieved the different burning conditions by varying the amount of air in the stoves, therefore changing the combustion temperature. For Set1, closing the air window decreased the flame temperature, resulting in a transition from a flaming to a smoldering fire. This could be visibly identified, together with the development of a thick white smoke from the chimney. We note that this conduct is very common in residential stoves, to keep the fire running for longer. Meanwhile, for set2, after lighting the fire, we kept a high air input to maintain a flaming fire. At the same time, we monitored the MCE in real time and only injected the emissions into the chamber when the MCE increased above 0.9. This information is now added to the manuscript, in Section 2.1:

*"Two smog chamber campaigns were conducted to investigate SOA production from multiple domestic wood combustion appliances as a function of combustion phase, initial fuel load, and OH exposure. These experiments were previously described in detail (Bruns et al., 2016; Ciarelli et al., 2017; Bertrand et al., 2017, 2018a) and are summarized here. Experiments from Bertrand et al. (2017, 2018a) will be referred to as Set1 and experiments from Bruns et al. (2016) and Ciarelli et al. (2017) as Set2. The emissions were generated by three different logwood stoves for residential wood combustion: stove 1 manufactured before 2002 (Cheminées Gaudin Ecochauff 625), stove 2 fabricated in 2010 (Invicta Remilly) and stove 3 (Avant, 2009, Attika). For each stove three to four replicate experiments were performed with a loading of 2-3 kg of beech wood having a total moisture content ranging between 2 and 19%. The fire was ignited with 3 starters made of wood wax, wood shavings, paraffin and 5 natural resin. The starting phase was not studied. In total, 14 experiments were performed, consisting of two experiments at -10°C, seven experiments at 2°C and five experiments at 15°C. These experiments cover the typical range of European winter temperatures and are summarized in Table 1.*

*Ward and Hardy (1991) define the flaming and smoldering conditions according to the modified combustion efficiency, MCE = $CO_2$/(CO+$CO_2$). Specifically, MCE > 0.9 is identified as flaming condition, while MCE < 0.85 is identified as smoldering condition. MCE values for the different experiments are reported in Table 1. According to this parameter, Set1 and Set2 experiments were dominated by smoldering and flaming, respectively. Practically, we achieved the different burning conditions by varying the amount of air in the stoves, therefore changing the combustion temperature. For Set1, closing the air window decreased the flame temperature, resulting in a transition from a flaming to a smoldering fire. This could be visibly identified, together with the development of a thick white smoke from the chimney. We note that this conduct is very common in residential stoves, to keep the fire running for longer. Meanwhile, for Set2, after lighting the fire, we kept a high air input to maintain a flaming fire. At the same time, we monitored the MCE in real time and only injected the emissions into the chamber when the MCE increased above 0.9."*

Abstract: From the abstract it appears that one of the major conclusions is that flaming and smoldering conditions produce a different mix of VOCs, but this is not discussed in the paper.

Figure 2, 3 and 4 aim to describe the different mix of primary VOCs for the two datasets, their chemical differences among the two sets of experiments, their contribution to the total primary emissions and the amount decayed upon oxidation.

Page 4 Line 1: These values don't quite make sense to me; how is it possible that the contributions consistently add to more than 100%? Is this mass yield or carbon yield?
The percentage values (84-116%) reported in the text are not SOA yields, but refer to what extent the measured SOA could be explained by the measured precursors based on their SOA mass yields reported in literature. For more information we refer to Bruns et al. (2016).

Page 4 Line 9: Which parameters?
The model parameters refer to oxidation product production rates and volatility distribution. For clarity, we modified the text as follows:
*"Based on the same experiments, the lumped concentrations of the 22 non-traditional volatile organic compounds and 2 lumped compound classes were constrained in a box model (Ciarelli et al., 2017a). Improved parameters were retrieved describing the volatility distributions and the production rates of the oxidation products from the overall mixture of precursors present in biomass smoke."*

Page 5 line 11: If relative humidity was constant, than actual water vapor mixing ratios were quite different between the three temperature conditions. Is this expected to have an effect? Why was this particular humidity condition chosen?
For the particle phase, the particle water content, which affects the partitioning of semi-volatile products, is a function of RH and not of absolute humidity. Thus, we kept RH constant in the experiments in order to compare the different systems. Regarding the gas phase chemistry, water concentration would play a role if the main source of OH radicals were ozone. However, in our case, OH radical is produced from HONO photo dissociation. Therefore we do not expect absolute humidity to play a significant role under our conditions. All experiments were conducted at an RH of ~ 50%, typical of daytime values encountered in Europe during winter.

Page 5: What NOx levels and NOx:VOC ratios were present?
NOx concentrations at the beginning of the experiments are available just for Set1, ranging between 0.04 and 0.26 ppm before lights on. Therefore, the NOx/VOCs levels varied between 0.1 and 0.5 and are therefore representative of high $NO_X$ conditions. These details were addressed in the previous publications.

Page 6 line 15: AMS collection efficiencies can be substantially less than 1, especially for very low volatility and highly oxidized particles. A collection efficiency correction could change the conclusions of this work. Can you support the assumption of CE=1?
The AMS measurements were thoroughly validated in the previous publications, reporting the results of the same experiments (Bruns et al., 2016; Bertrand et al., 2017; Kumar et al., 2018). Within the measurement uncertainties (20%), the sum of BC and non-refractory species is consistent with the

SMPS volume measurements when applying a CE of 1 and a density of 1.5, typical of biomass burning aerosols (Corbin et al., 2015).

Page 8 line 27: Was photolysis of VOCs considered?

Photolysis was not considered. We demonstrated that the decay of at least the major precursors is consistent with their loss by dilution and reaction with OH, indicating that other processes, including photolysis, are minor. For the other species, if photolysis occurs, it will have the same magnitude in both Set1 and Set2.

Page 9 Line 7: How were these 263 ions selected?

The 263 ions refer to all the species fitted using high-resolution analysis of the PTR data while the selected 86 compounds were selected as SOA precursors according to their clear decay upon oxidation in the smog chamber.

Eq. 4 and 5: Could the measured OA be simultaneously corrected for wall loss and dilution by dividing by the measured BC signal over time, normalized to 1 at t=0?

No, this is not possible. The loss of the particles onto the walls occurs through their deposition by diffusion and sedimentation and therefore affects only the particle phase. Accordingly, particle wall losses obey a first order process and therefore were corrected for using Eq. 5 and the wall loss rates determined from Eq. 4. In contrast, dilution does not only influence the physical removal of the particles, but also their partitioning between the gas and particle phase, which is dependent on the species volatility. Therefore, dilution cannot be simply corrected by normalizing to the BC concentrations. Instead, we have calculated the diluted concentrations of total condensable gases, which we use as input into the model to calculate the fractions that partition to the particle phase. The modelled particle phase after dilution is then compared with the measured wall loss corrected organic aerosol after dilution.

Page 11 Lines 30-Page 12 line 9: This needs to be better supported and more detailed. Each of the six main precursor classes was further subdivided into six volatility bins, and the average chemical properties were determined for each bin – is this correct? Was there a large range in #C, #H, #O within each bin? It is not entirely clear to me how the volatility bins were determined: was the volatility of each individual species determined, and species were then grouped into bins? Or were species first grouped using some other method, then the volatility of each group was determined?

Here, we believe that some clarifications are needed. The section refers to the precursor oxidation products, referred to as secondary surrogates, and not to the precursors themselves. The volatility of the precursor classes was not used in the model. The volatility basis set was used to distribute the oxidation products from the different precursor classes into logarithmically spaced volatility bins. These bins are preset; 6 bins were considered with a saturation concentration $C^* = \{10^{-1}; 10^0; 10^1; 10^2; 10^3; 10^4\}$ µg m$^{-3}$ for the oxidation products of all precursor classes. This is described in the model inputs section 3.3.2 in lines 26-28. The box model using a genetic algorithm as a solver determines the fraction of the oxidation products in each of the bins, or what can be referred to as the volatility distribution of the oxidation products. The molecular composition of the secondary surrogates, which is used to calculate

their molecular weight and the bulk OA O:C ratios, is based on that of their precursors, as described on page 13 Lines 6 and 13.

Page 13 lines 1-7: Can you explain more clearly what mutation, crossover, and selection mean for the implementation of the genetic algorithm in this particular application? How does the algorithm actually identify the optimal set of parameters?

Based on the reviewer comment, we rephrased Section 3.3.4 (on Page 13 Lines 1-7) as follows:

*"The model is optimized to determine the volatility distributions of the oxidation products from different precursor classes described by $\mu_j$, $\sigma$ and their temperature-dependence described by $\Delta H_{vap}$, to best fit the observed OA concentrations. For the model optimization, we used a genetic algorithm (GA), a metaheuristic procedure inspired by the theory of natural selection in biology, including selection, crossover and mutation processes, to efficiently generate high-quality solutions to optimize problems (Goldberg et al., 2007; Mitchell, 1996). The GA is initiated with a population of randomly selected individual solutions. The performance of each of these solutions is evaluated by a fitness function, and the fitness values are used to select more optimized solutions, referred to as parents. The new generation of solutions (denoted children) are produced either by randomly changing a single parent (as mutation) or by combing the vector entries of a pair of parents (as crossover). The evolution process will be repeated until the termination criterion is reached, here maximum iteration time. In this study, a population of 50 different sets of model parameters ( $\mu_j$, $\sigma$ and $\Delta H_{vap}$) was considered for each GA generation. The sum of mean bias and RMSE between measured and modelled $C_{OA}$ of the 14 experiments were used as fitness function to evaluate the solutions. We assume the termination criterion is reached if no improvement in the fitness occurs after 50 generations, with a maximum of 500 total iterations allowed. The GA calculations were performed using the package "GA" for R (Scrucca et al., 2017). A bootstrap method was then adopted to quantify the uncertainty in the constrained parameters."*

Page 14 lines 21-23: It should be stated more clearly before this point that the individual species contributing to each of the 6 chemical classes were not consistent between different experiments. The inconsistency in chemical composition makes it extremely difficult to compare between different experiments or to draw conclusions about the SOA yield from different chemical classes.

As mentioned above, the individual species included in each chemical class are the same for each experiment and for both datasets.

Page 15 lines 4-6: Two groups of compounds could have substantially different distribution of OH reactivities and SOA formation potential, but similar average OH reactivity. The conclusion here is not well supported.

The aim of this analysis is to investigate whether the faster decay observed against the OH exposure in Set1 compared to Set2 is due to the presence of more reactive species in Set1 for the different chemical classes. Therefore, in Table 2, we calculated average OH reaction rate constants for each single experiment and for the different chemical classes. These were weighted by the abundance of the different compounds in their respective chemical class. We show that the reaction rates compare well between the two sets of experiments, which does not explain why some chemical classes decay faster in set1 compared to set2.

**Technical corrections:**

Abstract 8-10: Sentence is difficult to understand, please rephrase.
Abstract was adjusted and is reported above.

Page 5 line 49: should be sodium nitrite, NaNO2
Yes, we apologize for this mistake. This was corrected in the revised manuscript.

Eq. 1 Brackets for the sum term are missing.
This was adjusted in the modified text.

*References:*

[revised manuscript text omitted]

---

## Author Comment (AC2) · 20 Jun 2019

**Response to reviewer's comments:**

**Title:** Secondary organic aerosol formation from smoldering and flaming combustion of biomass: a box model parametrization based on volatility basis set

**Journal:** Atmospheric Chemistry and Physics

**Manuscript ID:** acp-2018-1308

Dear Editor,

We thank the reviewers for their comments. Our detailed point-by-point responses to the reviewers' comments (in black regular typeset) are provided in blue regular typeset and the revised text (highlighted in the main text) is in *grey italic typeset*.

**Reviewer 2:**

**Summary:**

This paper is an interesting study on secondary organic aerosol formation from biomass burning. The authors conducted 14 experiments under two burning conditions (flaming, smoldering and flaming) and with different types of stove. Emissions from the burned stoves were sampled and were aged via OH-oxidation reactions to investigate the secondary organic aerosol yields and the chemical properties of the oxidation products from smoldering and flaming. A box model and a genetic algorithm approach were used to quantify the contribution of the VOC oxidation products to SOA yield and to better explain the SOA formation process. However, many portions of the paper were made difficult to follow due to missing details. I recommend that this study be published but with minor revisions. I request that the authors consider the following points as they revise this manuscript:

**General comments:**

1/ As the title indicates, this paper should focus mainly on SOA formation from smoldering and flaming combustion of biomass. However, few information is given in that issue. The authors should better describe the burning conditions of smoldering and flaming by adding more details especially on how the authors reproduce experimentally the flaming and smoldering combustion.

The first reviewer also asked for more details about the combustion conditions. In the new version of the manuscript, we now describe how we achieved the two different burning conditions. The new Section 2.1 reads as follows:

*"Two smog chamber campaigns were conducted to investigate SOA production from multiple domestic wood combustion appliances as a function of combustion phase, initial fuel load, and OH exposure. These experiments were previously described in detail (Bruns et al., 2016; Ciarelli et al., 2017; Bertrand et al., 2017, 2018a) and are summarized here. Experiments from Bertrand et al. (2017, 2018a) will be referred to as Set1 and experiments from Bruns et al. (2016) and Ciarelli et al. (2017) as Set2. The emissions were generated by three different logwood stoves for residential wood combustion: stove 1 manufactured before 2002 (Cheminées Gaudin Ecochauff 625), stove 2 fabricated in 2010 (Invicta Remilly) and stove 3 (Avant, 2009, Attika). For each stove three to four replicate experiments were*

*performed with a loading of 2-3 kg of beech wood having a total moisture content ranging between 2 and 19%. The fire was ignited with 3 starters made of wood wax, wood shavings, paraffin and 5 natural resin. The starting phase was not studied. In total, 14 experiments were performed, consisting of two experiments at -10°C, seven experiments at 2°C and five experiments at 15°C. These experiments cover the typical range of European winter temperatures and are summarized in Table 1.*

*Ward and Hardy (1991) define the flaming and smoldering conditions according to the modified combustion efficiency, $MCE = CO_2/(CO+CO_2)$. Specifically, $MCE > 0.9$ is identified as flaming condition, while $MCE < 0.85$ is identified as smoldering condition. MCE values for the different experiments are reported in Table 1. According to this parameter, Set1 and Set2 experiments were dominated by smoldering and flaming, respectively. Practically, we achieved the different burning conditions by varying the amount of air in the stoves, therefore changing the combustion temperature. For Set1, closing the air window decreased the flame temperature, resulting in a transition from a flaming to a smoldering fire. This could be visibly identified, together with the development of a thick white smoke from the chimney. We note that this conduct is very common in residential stoves, to keep the fire running for longer. Meanwhile, for Set2, after lighting the fire, we kept a high air input to maintain a flaming fire. At the same time, we monitored the MCE in real time and only injected the emissions into the chamber when the MCE increased above 0.9.''*

2/ According to Majdi et al. (2019), Koo et al. (2014), Konovalov et al. (2015) and Ciarelli et al. (2017), Intermediate and Semi Volatile Organic Compounds (I-SVOCs) are considered as one of the most important SOA precursors emitted by biomass burning. Why did the authors focus only on SOA from VOCs ?

There is a general inconsistency in the nomenclature in the literature. The studies cited above are mainly modelling studies, where the precursors are not measured, but are constrained in the model to improve its agreement with the measurements (Majdi et al. 2019, Koo et al. 2014, Konovalov et al. 2015). These authors refer to these precursors as I/SVOCs as it has been traditionally the case. While unidentified non-traditional precursors are indeed important for SOA formation from biomass burning, claims about the volatility of these precursors cannot be supported without measurements. We prefer to term these missing compounds as non-traditional precursors, without referring to their volatility, which in any case is not required in the model. Recently, Bruns et al. (2016) identified the main SOA precursors in flaming biomass, using a proton transfer reaction mass spectrometer (PTR-MS). These consist of mainly IVOC and VOCs, which have not been traditionally considered in the models. Ciarelli et al. (2017) used the same data to constrain the amounts of non-traditional precursors (the authors termed them non-traditional VOCs, NT-VOC) in a VBS model. They show that while the oxidation of primary SVOCs takes place, increasing the bulk O:C ratio, the net amount of mass gained from these precursors is not substantial (i.e. POA before aging compares well to the remaining POA + oxidized SVOCs after aging). Meanwhile, the oxidation of NT-VOCs are responsible for the majority of the observed enhancement in the OA mass, consistent with the results of Bruns et al. (2016). Based on the reviewer comment and to avoid any confusion, we will replace the term VOCs with OGs (organic gases) in the corrected version of the manuscript.

**Specific comments:**

1/ Page 4, lines 31-32: Why did the authors choose these experiments? How could the authors study the effect of smoldering/flaming combustion of biomass when the other experimental parameters (OH exposure, temperature of the smog chamber, stove.... ) change at the same time ?

Extracting class specific yields essentially resides on the variability in the contributions of different classes to condensable species, between experiments (due to variability in the emissions) or within an experiment (due to differences in the precursors reaction rates). By nature, emissions are very complex and therefore hard to control and typically several parameters co-vary. It is important in fitting for the class specific yields in such a multivariate problem that the contributions of the different precursors to the condensable gases do not co-vary with other parameters (e.g. temperature, OH exposure), otherwise their respective effects cannot be distinguished and biases in the yields may arise. The main variability in the contribution of different classes to condensable species is due to variability in the emissions between Set1 and Set2. Most of the other parameters in Set1 and Set2 are consistent or have a significant overlap, allowing the retrieval of class specific yields and a direct comparison between the experiments. In the following, we compare Set1 and Set2 with respect to the different combustion and aging parameters:

➔ Combustion conditions: The combustion conditions between the two sets are comparable, in terms of the starting procedure and the type and the amount of wood burned. The main difference is related to the air input, which favored smoldering and flaming conditions for Set1 and Set2, respectively. This is directly reflected in the composition of the SOA precursors and such variability is key to deconvolve the class specific yields. Despite this, the emission composition for Set2 is variable, with some experiments showing a similar composition as Set1, ensuring an overlap between Set1 and Set2 chemical composition when other parameters varied.

➔ OH exposure: The OH exposure is ~7 times higher for Set2 compared to Set1. This is because the OH production rate was similar between the two experiments, but its sink was significantly higher for Set1 due to the injection of higher emissions. However, we note that the OH exposure is not directly considered in the model. Instead, the integrated production of condensable species, inferred from the precursor decay, is used as inputs, which takes into account the differences in the OH exposures. Besides, we note that most of the SOA production occurs at OH exposures $< 8 \cdot 10^6$ molecules cm$^{-3}$ h, which is reached in both experiments.

➔ Emission loading: On average, higher concentrations were injected during Set1 compared to Set2. This was not intentional, but occurred as smoldering emissions rates were much more important and therefore it was more challenging to control the amounts injected. That said it is important that the concentrations span a wide range (here 1.5 orders of magnitude) to be able to constrain the amounts of oxidation products in the different volatility bins. As mentioned in the manuscript, the volatility distributions are well constrained in the saturation concentrations between 10-1000 µg m$^{-3}$, corresponding to the range of OA concentrations studied here. We finally note that the concentration ranges in Set1 and Set2 have a significant overlap (for Set1, SOA = 50-800 µg m$^{-3}$ vs. 20-140 µg m$^{-3}$), ensuring that the fitting is not biased due to difference in the loadings between the two sets.

➔ Aging temperature: conducting the experiments at different temperatures is essential for constraining the condensable species enthalpy of evaporation. Set1 experiments were conducted at 2°C, right in the middle of the two temperatures examined in Set2: -10°C and 15°C. Therefore, the temperature does not co-vary with the changes in the emission composition between Set1 and Set2 and the retrieval of the yields is not affected by the temperature.

To compare the SOA produced from smoldering and flaming emissions, we used the specific yields retrieved from box modeling and varied the initial emission composition keeping every other parameter constant. This is presented in Figures 10 and 11.

2/ Page 5, line 27: Flaming combustion occurs at high temperature. Can the authors give an order of magnitude of this high temperature ?
For flaming combustion, the temperature inside the stove was around 600°C.

3/ Page 5, line 31: Why did the authors choose to combine flaming and smoldering emissions in set 1?
We now describe clearly how the stoves were operated and how the different burning conditions were achieved. Practically, we achieved the different burning conditions by varying the amount of air in the stoves, therefore changing the combustion temperature. For Set1, closing the air window decreased the flame temperature, resulting in a transition from a flaming to a smoldering fire. This was identified visibly, together with the observation of a thick white smoke from the chimney. Here, we cannot exclude that flaming and smoldering occurred at the same time in the stove (we still observed some flames, but much less vivid). Therefore, we referred to Set1 as smoldering dominated. This is supported by the values of MCE~0.85. Meanwhile, for Set2, after lighting the fire, we kept a high air input to maintain a flaming fire. At the same time, we monitored the MCE in real time and only injected the emissions in the chamber when the MCE increased above 0.9.

4/ Page 7, line 7: What did the authors mean by "other processes"?
Other oxidative processes (e.g. $NO_3$ chemistry) or any other process that could deplete VOC concentrations upon aging.

5/ Page 9, line 6: Can the authors give more information about this common set of 263 ions used to identify the most important SOA precursors?
The 263 ions refer to all the HR fitted m/z values from the PTR analysis while the selected 86 compounds were selected as SOA precursors according to their clear decay upon oxidation in the smog chamber. The major SOA precursors amongst these 86 compounds were described in Bruns et al. (2016).

6/ Page 11, line 22-23: Why did the authors choose to set the number of volatility bins to 6 ? How was the volatility C* determined ? Did the authors measure the volatility of each species ?
The volatility basis set (VBS) scheme was used to distribute the oxidation products from the different precursor classes into logarithmically spaced volatility bins. These bins are preset; 6 bins were considered with a saturation concentration C* = {$10^{-1}$; $10^0$; $10^1$; $10^2$; $10^3$; $10^4$} µg m$^{-3}$ for the oxidation products of all precursor classes. These bins are chosen to cover the range of OA concentrations measured in our study (higher volatility compounds are not expected to contribute at our conditions). The box model using a genetic algorithm as a solver determines the fraction of the oxidation products in each of the bins, or what can be referred to as the volatility distribution of the oxidation products, to best fit the measured OA concentration. This is thoroughly described in Section 3.3.

7/ Page 11: More clarification is needed in section 3.3.2. Did the given chemical properties of surrogates represent the average of the compounds that are classified in a determined volatility bin ?

We did not measure the condensing compounds in the different volatility bins arising from the oxidation of the different precursor classes. As mentioned in the comment before, the amounts of these compounds are fitted to best represent the measured OA during the different experiments. The molecular composition of these compounds, needed to calculate their molecular weight, is inferred from that of their precursors (C# and H#).

8/ Page 12, line 11: How did the authors assume this single ΔH value for all surrogates? Did the authors consider any assumptions to determine this value?

The ΔH is one of the eight free parameters in the model along with the parameters describing the volatility distribution kernels. Therefore, ΔH is not assumed but fitted to best explain the temperature dependence of the yields. This is described in the modified Section 3.3.4 as follows:

*"The model is optimized to determine the volatility distributions of the oxidation products from different precursor classes described by $\mu_j$, $\sigma$ and their temperature-dependence described by $\Delta H_{vap}$, to best fit the observed OA concentrations."*

9/ Page 13, section 3.3.4 is very short. Can the authors clarify how did this algorithm work to identify the optimized parameters and what did the authors mean by the process of "natural selection"?

We rephrased Section 3.3.4 (in Page 14 Lines 1-15), adding more details about the genetic algorithm.

*"The model is optimized to determine the volatility distributions of the oxidation products from different precursor classes described by $\mu_j$, $\sigma$ and their temperature-dependence described by $\Delta H_{vap}$, to best fit the observed OA concentrations. For the model optimization, we used a genetic algorithm (GA), a metaheuristic procedure inspired by the theory of natural selection in biology, including selection, crossover and mutation processes, to efficiently generate high-quality solutions to optimize problems (Goldberg et al., 2007; Mitchell, 1996). The GA is initiated with a population of randomly selected individual solutions. The performance of each of these solutions is evaluated by a fitness function, and the fitness values are used to select more optimized solutions, referred to as parents. The new generation of solutions (denoted children) are produced either by randomly changing a single parent (as mutation) or by combing the vector entries of a pair of parents (as crossover). The evolution process will be repeated until the termination criterion is reached, here maximum iteration time. In this study, a population of 50 different sets of model parameters ($\mu_j$, $\sigma$ and $\Delta H_{vap}$) was considered for each GA generation. The sum of mean bias and RMSE between measured and modelled $C_{OA}$ of the 14 experiments were used as fitness function to evaluate the solutions. We assume the termination criterion is reached if no improvement in the fitness occurs after 50 generations, with a maximum of 500 total iterations allowed. The GA calculations were performed using the package "GA" for R (Scrucca et al., 2017). A bootstrap method was then adopted to quantify the uncertainty in the constrained parameters."*

10/ Page 13, Figure 2: The 'others' VOC emissions show the highest relative contribution mainly in set 1. What are these 'others' VOCs emissions?

'Others' include the class of species below *m/z* 66, which comprises compounds that do not show a clear decay upon oxidation and are therefore not considered as SOA precursors in the further analysis. The major contributor is acetic acid, previously reported to be dominant in residential wood burning emissions (Bhattu et al., 2019). We have this information in the modified version of the manuscript (Page 13, line 17):

*"The chemical class referred to as 'Others' comprise compounds that do not show a clear decay upon oxidation and are therefore not considered as SOA precursors in the following analysis. 'Others' is dominated by acetic acid, previously reported as a major species in residential wood burning emissions (Bhattu et al., 2019)."*

11/ Page 14, line 11: "Relevant compounds in the latter class are benzene, toluene and xylene." According to Bruns et al. (2016) experimental SOA yields, benzene is the third principal contributor to SOA after phenol and naphthalene. Did the authors characterize these SAH compounds?
The SAH are included in the study and contains benzene, toluene and xylene (table S1). These species exhibit higher contribution to the primary emissions for the flaming phase set of experiments compared to the smoldering phase.

12/ Page 14, line 20: "The SOA/POA ranges between 2 and 6, similar to ratios observed in previous studies." Please add references.
Following the reviewers comment, we added in the main text the following references: (Heringa et al., 2011a, Bruns et al., 2015, Grieshop et al., 2009a, Tiitta et al., 2016).

13/ Page 14, lines 22-23: "Such inconstancy in behavior is either due to differences in the chemical composition within the same class." How did the authors defined these chemical compositions? And how can these compositions change within the same class? Can this be explained by differences in the burned stoves? How can the authors compare SOA contribution from the different classes if the chemical composition can change? Please explain this further.
The chemical compounds considered in each class are consistent among datasets and single experiments. However, the relative abundance of these species within the same class may vary, although this variability is not significant. We calculated an average $k_{OH}$ per class for each experiment (Table 2), to show that the reaction rate within the same class of compounds is consistent among experiments and datasets.

14/ Page 18, lines 17-18: "In general for both phases studied, higher contributions to SOA formation was found for cresol and phenol species and chemically similar compounds." This conclusion is not well supported and not discussed in the paper. Please clarify or add a reference.
Cresol, phenol and chemically similar species belong to the family called OxyAH (see table S1). Figure 11 shows the fractional contribution to SOA for each family for the different Sets as a function of OH exposure, temperature and initial OM load. Overall the higher contribution for both phases investigated belongs to the OxyAH family. We modified this sentence to better relate it to the Figure 11:

*"For both phases, SOA formation is found to be dominated by OxyAH (e.g. phenols and cresols), emitted from lignin pyrolysis."*

**Technical comments:**

1)      Figure 7 caption: Please replace "the model tend to underestimate ... "by "the model tends to underestimate...".
We replaced 'tend' by 'tends'.

2)      Eq (1) page 7: Is there any missing bracket in the equation ? Please verify.
Yes, we apologize for this. The bracket was added in the equation.

3)      Figure 2 caption: Did Set1 represent only smoldering phase or smoldering and flaming as defined in page 5 line 31? Please correct.
Set1 represents smoldering dominated combustion and Set2 represents flaming dominated combustion. We cannot rule out the possibility of partial inclusion of the other type of combustion because of the challenging operation of the stoves investigated.

*References:*

Bertrand, A., Stefenelli, G., Bruns, E. A., Pieber, S. M., Temime-Roussel, B., Slowik, J. G., Prévôt, A. S. H., Wortham, H., El Haddad, I. and Marchand, N.: Primary emissions and secondary aerosol production potential from woodstoves for residential heating: Influence of the stove technology and combustion efficiency, Atmos. Environ., 169, 65–79, doi:10.1016/j.atmosenv.2017.09.005, 2017.

Bertrand, A., Stefenelli, G., Jen, C. N., Pieber, S. M., Bruns, E. A., Ni, H., Temime-Roussel, B., Slowik, J. G., Goldstein, A. H., El Haddad, I., Baltensperger, U., Prévôt, A. S. H., Wortham, H. and Marchand, N.: Evolution of the chemical fingerprint of biomass burning organic aerosol during aging, Atmos. Chem. Phys., 18(10), 7607–7624, doi:10.5194/acp-18-7607-2018, 2018a.

Bhattu, D., Zotter, P., Zhou, J., Stefenelli, G., Klein, F., Bertrand, A., Temime-Roussel, B., Marchand, N., Slowik, J. G., Baltensperger, U., Prévôt, A. S. H., Nussbaumer, T., El Haddad, I. and Dommen, J.: Effect of stove technology and combustion conditions on gas and particulate emissions from residential biomass combustion, Env. Sci Technol, 53(4), 2209–2219, doi:10.1021/acs.est.8b05020, 2019.

Bruns, E. A., El Haddad, I., Slowik, J. G., Kilic, D., Klein, F., Baltensperger, U. and Prévôt, A. S. H.: Identification of significant precursor gases of secondary organic aerosols from residential wood combustion, Sci. Rep-Uk, 6(1), doi:10.1038/srep27881, 2016.

Bruns, E. A., Krapf, M., Orasche, J., Huang, Y., Zimmermann, R., Drinovec, L., Močnik, G., El-Haddad, I., Slowik, J. G., Dommen, J., Baltensperger, U. and Prévôt, A. S. H.: Characterization of primary and secondary wood combustion products generated under different burner loads, Atmospheric Chem. Phys., 15(5), 2825–2841, doi:10.5194/acp-15-2825-2015, 2015.

Ciarelli, G., Aksoyoglu, S., El Haddad, I., Bruns, E. A., Crippa, M., Poulain, L., Äijälä, M., Carbone, S., Freney, E., O'Dowd, C., Baltensperger, U. and Prévôt, A. S. H.: Modelling winter organic aerosol at the European scale with CAMx: evaluation and source apportionment with a VBS parameterization based on novel wood burning smog chamber experiments, Atmos. Chem. Phys., 17(12), 7653–7669, doi:10.5194/acp-17-7653-2017, 2017b.

Ciarelli, G., El Haddad, I., Bruns, E., Aksoyoglu, S., Möhler, O., Baltensperger, U. and Prévôt, A. S. H.: Constraining a hybrid volatility basis-set model for aging of wood-burning emissions using smog

chamber experiments: a box-model study based on the VBS scheme of the CAMx model (v5.40), Geosci. Model Dev., 10(6), 2303–2320, doi:10.5194/gmd-10-2303-2017, 2017a.

Goldberg, D. E., Sastry, K. and Llorà, X.: Toward routine billion-variable optimization using genetic algorithms, Complexity, 12(3), 27–29, doi:10.1002/cplx.20168, 2007.

Grieshop, A. P., Logue, J. M., Donahue, N. M. and Robinson, A. L.: Laboratory investigation of photochemical oxidation of organic aerosol from wood fires 1: measurement and simulation of organic aerosol evolution, Atmospheric Chem. Phys., 9(4), 1263–1277, doi:10.5194/acp-9-1263-2009, 2009.

Heringa, M. F., DeCarlo, P. F., Chirico, R., Tritscher, T., Dommen, J., Weingartner, E., Richter, R., Wehrle, G., Prévôt, A. S. H. and Baltensperger, U.: Investigations of primary and secondary particulate matter of different wood combustion appliances with a high-resolution time-of-flight aerosol mass spectrometer, Atmospheric Chem. Phys., 11(12), 5945–5957, doi:10.5194/acp-11-5945-2011, 2011.

Konovalov, I. B., Beekmann, M., Berezin, E. V., Petetin, H., Mielonen, T., Kuznetsova, I. N. and Andreae, M. O.: The role of semi-volatile organic compounds in the mesoscale evolution of biomass burning aerosol: a modeling case study of the 2010 mega-fire event in Russia, Atmospheric Chem. Phys., 15(23), 13269–13297, doi:10.5194/acp-15-13269-2015, 2015.

Koo, B., Knipping, E. and Yarwood, G.: 1.5-Dimensional volatility basis set approach for modeling organic aerosol in CAMx and CMAQ, Atmos. Environ., 95, 158–164, doi:10.1016/j.atmosenv.2014.06.031, 2014.

Majdi, M., Turquety, S., Sartelet, K., Legorgeu, C., Menut, L. and Kim, Y.: Impact of wildfires on particulate matter in the Euro-Mediterranean in 2007: sensitivity to some parameterizations of emissions in air quality models, Atmospheric Chem. Phys., 19(2), 785–812, doi:10.5194/acp-19-785-2019, 2019.

Mitchell, M.: An introduction to genetic algorithms, Camb. Mass. Lond. Engl. Fifth Print., 3, 62–75, 1999.

Scrucca, L.: On some extensions to GA package: hybrid optimisation, parallelisation and islands evolution, arXiv preprint arXiv:1605.01931, 2017.

Tiitta, P., Leskinen, A., Hao, L., Yli-Pirilä, P., Kortelainen, M., Grigonyte, J., Tissari, J., Lamberg, H., Hartikainen, A., Kuuspalo, K., Kortelainen, A.-M., Virtanen, A., Lehtinen, K. E. J., Komppula, M., Pieber, S., Prévôt, A. S. H., Onasch, T. B., Worsnop, D. R., Czech, H., Zimmermann, R., Jokiniemi, J. and Sippula, O.: Transformation of logwood combustion emissions in a smog chamber: formation of secondary organic aerosol and changes in the primary organic aerosol upon daytime and nighttime aging, Atmospheric Chem. Phys., 16(20), 13251–13269, doi:10.5194/acp-16-13251-2016, 2016.

Ward, D. E. and Hardy, C. C.: Smoke emissions from wildland fires, Environ. Int., 17(2–3), 117–134, doi:10.1016/0160-4120(91)90095-8, 1991.